# Small molecule branched-chain ketoacid dehydrogenase kinase (BDK) inhibitors with opposing effects on BDK protein levels

Rachel J. Roth Flach [1,3] ✉, Eliza Bollinger[1,3], Allan R. Reyes[1], Brigitte Laforest[1], Bethany L. Kormos [1], Shenping Liu [2], Matthew R. Reese [2], Luis A. Martinez Alsina[2], Leanne Buzon[2], Yuan Zhang[2], Bruce Bechle[2], Amy Rosado[2], Parag V. Sahasrabudhe [2], John Knafels [2], Samit K. Bhattacharya[1], Kiyoyuki Omoto[1], John C. Stansfield[1], Liam D. Hurley[1], LouJin Song[1], Lina Luo [2], Susanne B. Breitkopf[1], Mara Monetti[1], Teresa Cunio [1], Brendan Tierney[2], Frank J. Geoly[2], Jake Delmore[1], C. Parker Siddall[1], Liang Xue [1], Ka N. Yip[1], Amit S. Kalgutkar [1], Russell A. Miller[1], Bei B. Zhang[1] & Kevin J. Filipski [1] ✉

Branched chain amino acid (BCAA) catabolic impairments have been implicated in several diseases. Branched chain ketoacid dehydrogenase (BCKDH) controls the rate limiting step in BCAA degradation, the activity of which is inhibited by BCKDH kinase (BDK)-mediated phosphorylation. Screening efforts to discover BDK inhibitors led to identification of thiophene PF-07208254, which improved cardiometabolic endpoints in mice. Structure-activity relationship studies led to identification of a thiazole series of BDK inhibitors; however, these inhibitors did not improve metabolism in mice upon chronic administration. While the thiophenes demonstrated sustained branched chain ketoacid (BCKA) lowering and reduced BDK protein levels, the thiazoles increased BCKAs and BDK protein levels. Thiazoles increased BDK proximity to BCKDH-E2, whereas thiophenes reduced BDK proximity to BCKDH-E2, which may promote BDK degradation. Thus, we describe two BDK inhibitor series that possess differing attributes regarding BDK degradation or stabilization and provide a mechanistic understanding of the desirable features of an effective BDK inhibitor.

Branched-chain amino acids (BCAAs) leucine (Leu), isoleucine (Ile), and valine (Val) are essential amino acids such that levels are tightly regulated to provide sufficient substrate for protein synthesis while avoiding high concentrations, which can cause toxicity[1,2]. BCAAs are catabolized first by reversible transamination by branched-chain amino transferase (BCAT) to corresponding branched-chain ketoacids (BCKAs) ketoleucine, ketoisoleucine, and ketovaline. The first committed step is BCKA decarboxylation by branched-chain ketoacid

dehydrogenase (BCKDH). Subsequent CoA species are further metabolized, and their terminal products acetyl CoA or succinyl CoA, are tricarboxylic acid (TCA) cycle substrates[3]. The BCKDH enzyme complex consists of E1, an α-ketoacid decarboxylase containing E1α (BCKDHA) and E1β (BCKDHB); E2, a dihydrolipoyl transacylase; and E3, a dihydrolipoamide dehydrogenase[3,4]. BCKA flux through the complex is inhibited by E1α phosphorylation (Ser337 in the full-length human protein) by branched-chain ketoacid dehydrogenase kinase (BDK or

---

[1]Pfizer Worldwide Research, Development & Medical, Cambridge, MA 02139, USA. [2]Pfizer Worldwide Research, Development & Medical, Groton, CT 06340, USA. [3]These authors contributed equally: Rachel J. Roth Flach, Eliza Bollinger. ✉e-mail: rachel.rothflach@pfizer.com; kevin.filipski@pfizer.com

BCKDK)[5]. The phosphatase PPM1k (aka PP2Cm) dephosphorylates this site to increase BCKA catabolism[6,7]. BCKDHE2 makes up the complex core, and BDK and PPM1k are thought to compete for E2 binding to regulate activity[8,9].

BCAA and/or BCKA levels are elevated in diseases including heart failure (HF)[10–13], type 2 diabetes mellitus (T2DM)[14,15], non-alcoholic fatty liver disease (NAFLD)[16], and obesity[17–19]. High BCAA levels are indicators for future T2DM[20] and pancreatic cancer[21] development. HF and T2DM are also associated with a loss of BCAA catabolic machinery[10,11], suggesting that BCKDH activity and/or BCAA flux could be dysregulated in these diseases. Therefore, increasing BCAA catabolism with a BDK inhibitor could be an effective treatment for cardiometabolic diseases.

BDK is an atypical kinase containing a regulatory domain with an allosteric inhibitory site where endogenous ketoleucine[8] as well as previously identified BDK inhibitors such as (S)-CPP, phenylbutyrate[8] BT2[4] and aryl-tetrazoles[22] bind. Clofibric acid[23] has an undetermined binding location, but may bind to this site as well based on structure similarity[8]. Thiamine pyrophosphate[24] has also been described as a BDK inhibitor; however, its direct interaction with BDK has not been established and furthermore is known to be a BCKDH cofactor. Herein, we identify the thiophene PF-07208254 as an allosteric BDK inhibitor with improved potency over BT2. PF-07208254 has a similar efficacy profile to BT2 and promotes BDK degradation, which contributes to sustained BCKA lowering in animals. An alternative thiazole BDK inhibitor series was identified with superior potency. However, the thiazoles did not sustainably improve metabolism in mice, and BCKA levels rebounded above baseline at low compound concentrations, suggesting differing pharmacology from PF-07208254 and BT2. Mechanistically, the thiazoles, while inhibiting BDK activity, promoted the BDK-E2 interaction and increased BDK protein levels, thus explaining the BCKA rebound observed and lack of sustained metabolic efficacy. Overall, these efforts resulted in identification of a class of BDK inhibitors that increase BDK protein levels. Our studies suggest that not only BDK inhibition but also BDK degradation is an important attribute of an effective BDK inhibitor.

## Results

To discover a potent and selective BDK inhibitor, a subset of the Pfizer compound collection was screened. Low molecular weight acids, compounds with high similarity scores to known inhibitors, and compounds that scored well in a virtual screen of the allosteric binding site (~12,000 compounds in total) were tested for BDK potency in a fluorescence resonance energy transfer (FRET)-based assay that measures BDK-induced phosphorylation of the E2 Receptor Binding Domain (RBD) covalently bound to a peptide sequence matching the E1 phosphorylation recognition motif (see Supplementary Information). Two hits, **S1** and **S2**, shared a common trans-dithiophene carboxylic acid chemotype and showed sub-micromolar potency (Supplementary Table 1). Through structure-activity relationship (SAR) optimization, PF-07208254 was identified (Table 1). A surface plasmon resonance (SPR)-based binding assay was developed with PF-07208254 showing good agreement between binding $K_d$ (84 nM) and functional $K_i$ (54 nM). Inhibitory potency was also determined in human skeletal myocytes using an AlphaLISA SureFire Ultra detection system to monitor BCKDH phosphorylation (cellular $IC_{50}$ = 540 nM). Off target activity was assessed in both CEREP and kinase panels (Supplementary Tables 2, 3).

An X-ray crystal structure of PF-07208254 bound to human BDK (Supplementary Figs. 1, 4A) revealed that PF-07208254 bound to the allosteric inhibitory pocket of the BDK regulatory domain similar to (S)-CPP and BT2[4,8]. The superior potency of PF-07208254 relative to BT2 may be due to improved protein contacts with non-polar side chains in the binding pocket or polarization of the sulfur electron density.

## PF-07208254 improves cardiac function and metabolism in mice

Previous studies showed that BT2 improved cardiac function in mouse HF models including transverse aortic constriction (TAC)[10,11,25–27]. To assess whether PF-07208254 also improved cardiac function in TAC, animals were administered PF-07208254 or BT2 in rodent chow and subjected to sham or TAC surgery (Fig. 1A). Body weight throughout the study was unchanged (Supplementary Fig. 2A), but food consumption was significantly reduced in the BT2-chow group (Supplementary Fig. 2B). Four weeks after TAC, echocardiography was performed, and a significant reduction in fractional shortening (FS%) and ejection fraction (EF%) was observed in TAC mice, and both PF-07208254 and BT2 improved FS% with a trend towards improvement in EF% (Fig. 1B, C). Heart rate was unaltered (Supplementary Fig. 2C), and other echocardiography parameters including volume at diastole, cardiac output, stroke volume, and left ventricular internal diameter in diastole (LVIDd) were not significantly altered by PF-07208254, while volume at diastole was significantly improved with BT2 (Supplementary Fig. 2D-G). At euthanasia, heart and lung weights were significantly increased in mice subjected to TAC. However, heart weight trended towards reduction by PF-07208254 and was significantly reduced by BT2 treatment, while lung weights trended to reduction with both compounds (Fig. 1D, E). pBckdh and Bdk protein levels were significantly reduced in heart by PF-07208254 and BT2 treatment (Fig. 1F-H). Finally, plasma BCAA/BCKAs were significantly reduced, and total circulating concentrations of PF-07208254 and BT2 were $23,899 \pm 14,153$ ng/mL and $216,000 \pm 19370$ ng/mL, respectively (Fig. 1I, J, Supplementary Fig. 2H-K). Thus, PF-07208254 and BT2 improve cardiac function and remodeling in a qualitatively and quantitatively similar manner.

BT2 has demonstrated metabolic improvements in diabetic rodent models[28–30], so metabolic parameters were also assessed after PF-07208254 administration. Mice that had been fed a high fat diet (HFD) for 10 weeks were administered vehicle, a low or high dose of PF-07208254, or BT2 daily for 8 weeks (Fig. 2A). Body weights were not altered by PF-07208254, although BT2-treated animals had reduced body weight (Supplementary Fig. 3A). An oral glucose tolerance test (oGTT) was performed at day 2 (Fig. 2B, C) and week 2 (Fig. 2D, E) 1 h post compound dose. Both PF-07208254 and BT2 administration significantly reduced glucose excursion by 10-14% at both day 2 and week 2 (Fig. 2B-E). Upon euthanasia, hepatic steatosis was measured biochemically and histologically by a certified veterinary pathologist (Fig. 2F, G). While liver weights were unchanged across the treatment groups (Supplementary Fig. 3B), triglyceride (TG) content and steatosis score was significantly reduced by BT2, and a strong trend to reduced TG content ($p = 0.06$) and a non-signficant reduction in steatosis score was observed with PF-07208254 (Fig. 2F, G). A reduction in inflammatory genes Ccl2, Ccr2, and Cd68 and fibrotic genes Col1a1 and Col1a2 was observed in liver with both PF-07208254 (90 mg/kg) and BT2 (Supplementary Fig. 3C). Fasting insulin levels were also reduced with PF-07208254 and BT2 (Fig. 2H), suggesting that PF-07208254 improves whole body metabolism in mice in a phenotypically similar manner to BT2.

After 7 weeks of treatment, a phamacokinetic/pharmacodynamic (PK/PD) experiment was conducted in which circulating levels of PF-07208254 were measured concurrent with BCAA/BCKA levels. Significant, dose- and time-dependent reductions in BCAA and BCKA were observed with PF-07208254 (Fig. 2I, J, Supplementary Fig. 3D–G). Interestingly, plasma BCKA levels remained significantly lower in PF-07208254-treated animals despite drug levels being below the lower limit of quantitation (LLOQ) (Supplementary Fig. 3H), which would not be expected to result in observable PD. In a separate study, after 8 days of PF-07208254 or BT2 treatment (Supplementary Fig. 3I), a significant, dose-dependent reduction in muscle BCAA/BCKA was observed (Supplementary Fig. 3J, K), similar to the observation of reduced plasma BCAA/BCKA levels. These data

**Table 1 | Potency of BDK inhibitors**

| Compound | Structure | BDK in vitro IC$_{50}$ (nM)[a] | BDK SPR K$_D$ (nM)[a] | Human skeletal myocyte IC$_{50}$ (nM)[a] |
|---|---|---|---|---|
| BT2 | | 1100 ± 27 | 490 ± 59 | 4100 ± 340 |
| PF-07208254 | | 110 ± 6.9 ($K_i$ = 54 ± 3.1) | 84 ± 8.7 | 540 ± 140 |
| PF-07238025 | | 4.5 ± 0.58 | 4.4 ± 1.1 | 59 ± 10 |
| PF-07247685 | | 0.86 ± 0.15 | 0.68 ± 0.38 | 3.0 ± 0.43 |

[a]Potency values are ± standard error with an N ≥ 3.

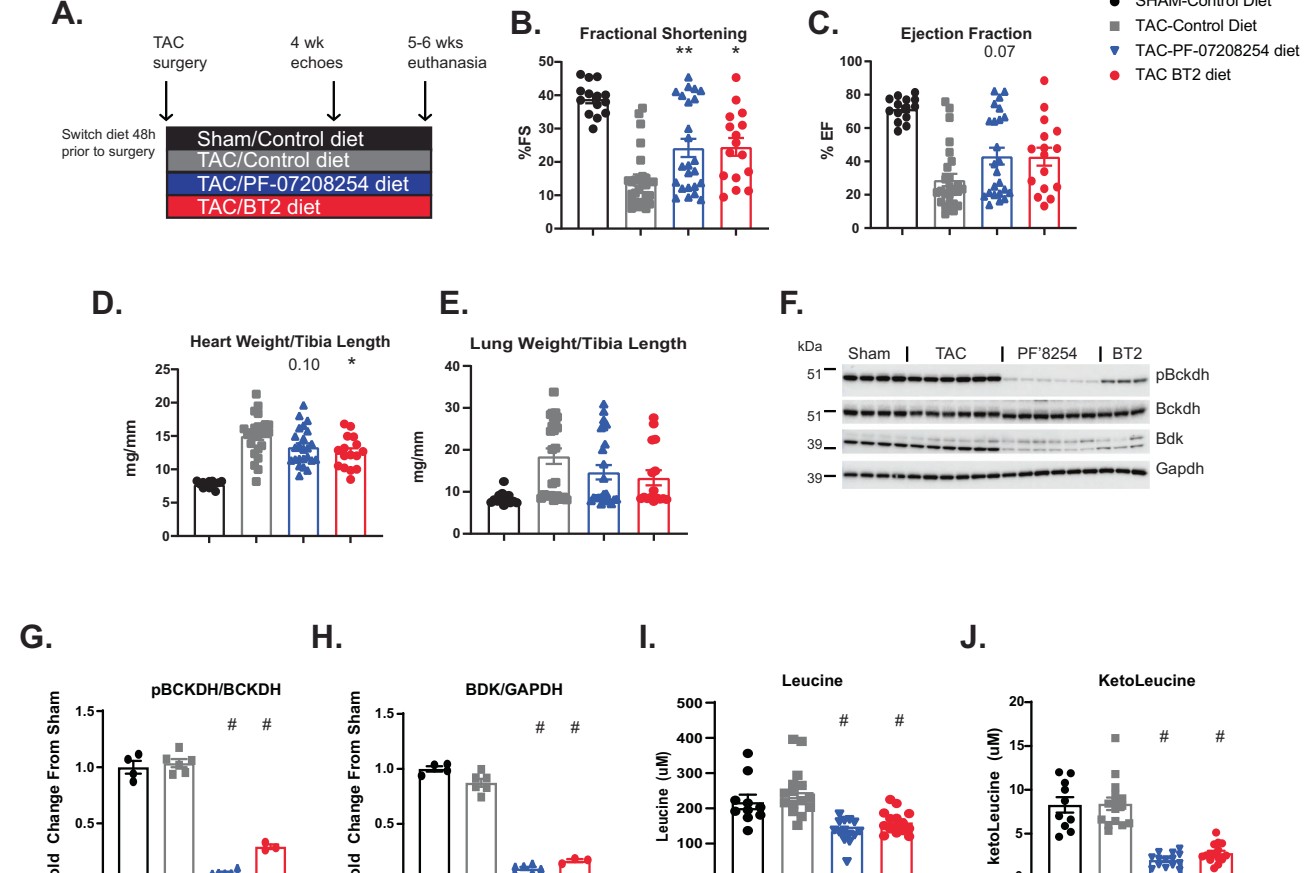

**Fig. 1 | PF-07208254 improves cardiac function similar to BT2 and reduces pBCKDH in mice. A**–**J** Mice were subjected to TAC surgery. **A** Study design. Mice were switched to chow containing PF-07208254 or BT2 48 h prior to surgery. Four weeks after surgery, echocardiography was performed. Five to six weeks after surgery, the animals were euthanized. **B**, **C** Echocardiography measurements **B**. % Fractional shortening (%FS) (**$p$ = 0.007, *$p$ = 0.015), **C** % Ejection fraction (%EF). **D**–**E** Tissue weights were measured at euthanasia. **D** Heart weight normalized to tibia length (*$p$ = 0.018). **E** Lung weight normalized to tibia length. ($N$ = 14–26 animals/group for **B**–**E**; statistics performed by one-way ANOVA with Tukey post-hoc test for **B**–**D** and pairwise Wilcoxon test was performed for **E**). **F**–**H** Heart tissue was immunoblotted. **F** Representative Western blot images of pBckdh, Bckdh, BDK and Gapdh. **G-H** Densitometry from **F** ($N$ = 3–6 animals/group; statistics performed by pairwise Wilcoxon test. #; $p < 0.0001$). **I**, **J** Plasma BCAA and BCKA levels were quantified at day 38. **I** Leucine. **J** Ketoleucine ($N$ = 10–15 animals/group; statistics performed by one-way ANOVA with Tukey post-hoc test. #$p$ = 0.0001). Data represent the mean ± SEM. Source data are provided as a Source Data file.

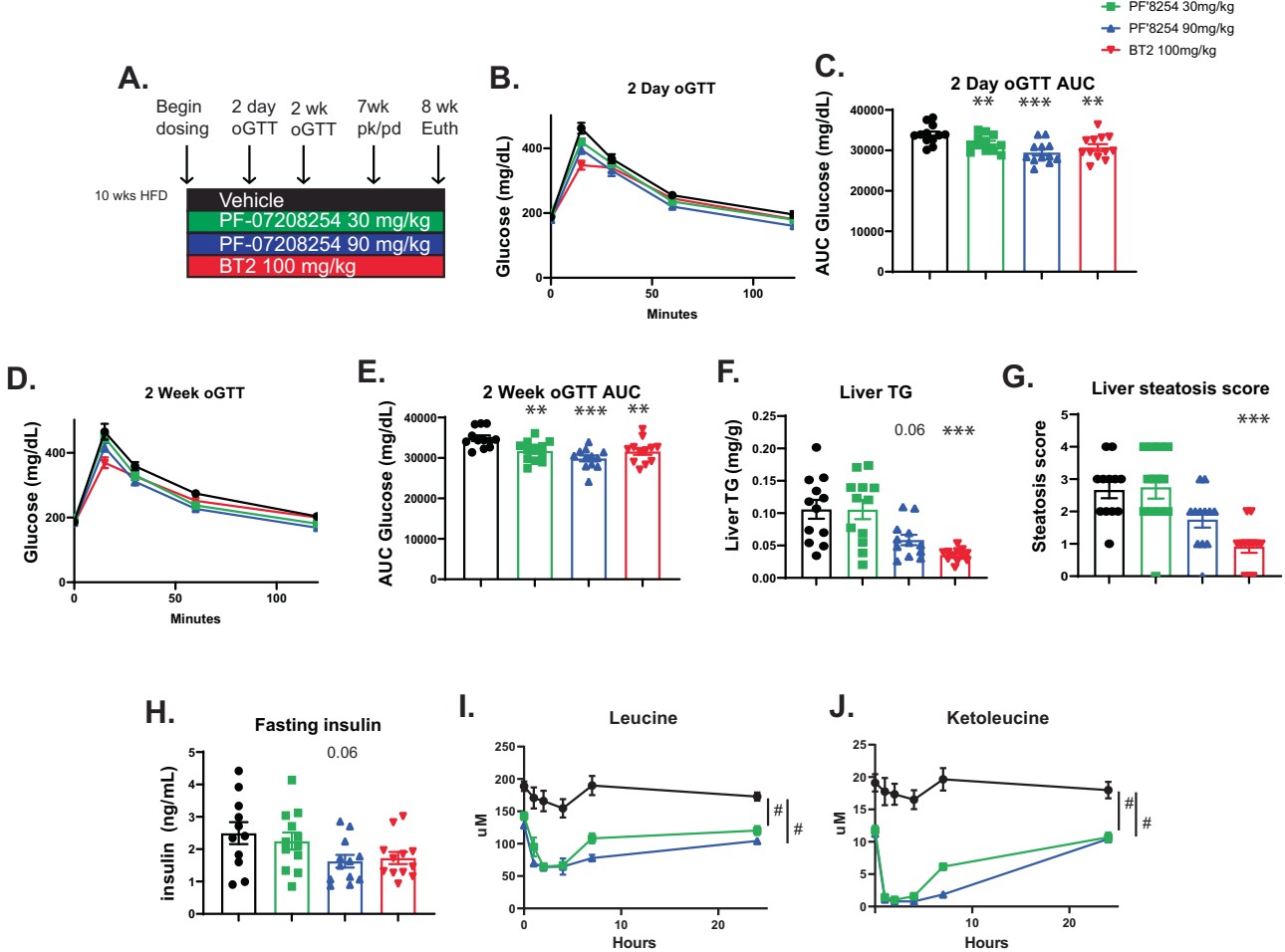

**Fig. 2 | PF-07208254 improves metabolism similar to BT2 and sustainably reduces BCAA and BCKA in mice.** Mice were fed HFD for 10 weeks, at which time animals were randomized into groups, and treated daily with vehicle, PF-07208254 or BT2. **A** Study design. **B–H** Oral glucose tolerance tests (oGTTs) were performed ($N = 12$ animals/group; a longitudinal mixed effects model with a random intercept and an AR(1) covariance structure was fit for each mouse for the glucose AUCs over the course of the study). **B** Day 2 oGTT. **C** Day 2 oGTT AUC. **D** Week 2 oGTT. **E** Week 2 oGTT AUC (**p = 0.009, ***p < 0.0001, **p = 0.009). **F–H** Plasma and livers were isolated 1 h post final compound dose, and steatosis was evaluated by histology ($N = 12$ animals/group). **F** Hepatic triglycerides (statistics performed with pairwise Wilcoxon test, ***p = 0.003). **G** Hepatic steatosis was graded by a veterinary pathologist on a scale of 0-4. (Statistics was performed using a Kruskal Wallis test followed by a Dunn test to compare within groups, ***p = 0.001). **H** Plasma insulin levels ($N = 11$–12 animals/group; statistics performed by one-way ANOVA). **I, J** A PK/PD assessment was performed after 7 weeks of treatment in which mice are dosed with compound, and timed bleeds were performed to measure drug levels, BCAA, and BCKA levels ($N = 6$–12 animals/group; statistics performed with one-way ANOVA with Tukey HSD test). **I** Leucine (#p < 0.0001), **J** Ketoleucine (#p < 0.0001). Data represent the mean ± SEM. Source data are provided as a Source Data file.

suggest that PF-07208254 improves BCAA catabolism and lowers BCAA/BCKA levels in mice. In summary, based upon its improved BDK inhibitory potency, BCKA/BCAA lowering, cardio-metabolic efficacy and other attributes, PF-07208254 was nominated as the first known BDK inhibitor development candidate.

### Discovery and pharmacology of thiazole BDK inhibitors

Additional SAR optimization was initiated on a less ligand efficient, structurally distinct screening hit, **S3**. **S3**, which was one of the initial hits found from our targeted file screening efforts detailed above, was intriguing because this weak inhibitor bound in a distinct mode adjacent to the allosteric binding site (Fig. 3A, Supplementary Fig. 4B). To accommodate **S3** binding, L128, R167 and R171 side chains adopt different rotamers than in the BT2 complex. The phenyl ring tethered to the carboxylic acid in **S3** extends towards the kinase domain, forming interactions with Y241 and Y346, which are not observed in the BT2/BDK structure. The carboxylate also forms charged interactions with the side chains of R167 and polar interactions with the side chains of

R171 and Y346 via water-mediated hydrogen bonds. The R171 side chain Pi-stacks with the phenyl ring containing the carboxylic acid, and the N atom of the central thiazole ring is also involved in a water-mediated hydrogen bond network.

The terminal methyl group of **S3** positioned at the cusp of the lipophilic pocket filled by the halo-dithiophene motif of PF-07208254 presented an opportunity to improve potency by filling this pocket with lipophilic groups. This work culminated in the discovery of two analogs, PF-07238025 (cellular $IC_{50} = 59$ nM) and PF-07247685 (cellular $IC_{50} = 3$ nM) with PF-07247685 having a cellular potency three orders of magnitude greater than BT2. The profound potency improvements over **S3** and previously known BDK inhibitors can be attributed to the addition of the branched alkyl chains present in both molecules. Figure 3B and Supplementary Fig. 4C show the X-ray crystal structure of PF-07247685 bound to BDK. The central thiazole and benzoic acid of PF-07247685 overlap with **S3** and make similar interactions, while the butyl cyclopropyl group extends into the hydrophobic pocket of the allosteric site and makes extensive hydrophobic interactions.

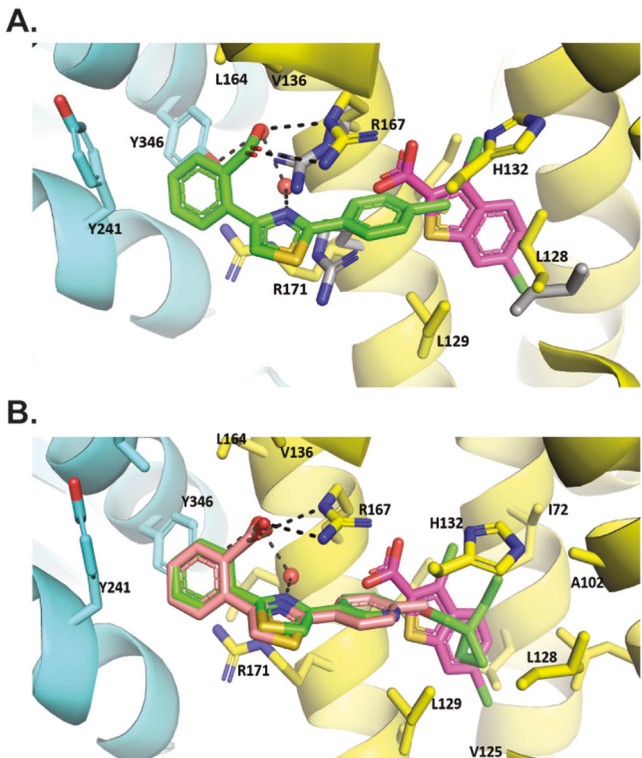

**Fig. 3 | Thiazoles bind to BDK in distinct mode adjacent to thiophenes.** Allosteric binding site from X-ray crystal structures of BDK bound to inhibitors. The regulatory domain of BDK is shown in yellow ribbons, and the kinase domain in aqua. Side chains of contacting residues are shown, and hydrogen bonds in dash. **A** S3 (stick model, C atoms colored in green) bound to hBDK. Superimposed is the BT2 X-ray crystal structure (PDB ID: 4E00) with BT2 C atoms colored in pink and side chains of residues that adopt different rotamers in the BT2 complex in gray. **B** PF-07247685 (stick model, C atoms colored in green) bound to hBDK. Superimposed are BT2 (C atoms colored in pink) and S3 (C atoms colored in salmon).

PF-07238025 and PF-07247685 were advanced to in vivo studies. First, HFD mice were treated with PF-07238025 for 9 days to assess PK/PD and glucose tolerance (Fig. 4A). In an oGTT performed on day 2, similar to PF-07208254, a significant improvement in glucose excursion was observed with PF-07238025 (Fig. 4B, C). PF-07238025 was next advanced to a chronic study (Fig. 4D), and an oGTT was performed after 2 weeks of treatment. However, PF-07238025 no longer improved glycemia (Fig. 4E, F), in contrast to PF-07208254 and BT2 (Fig. 2D, E).

Body weights were not significantly altered with PF-07238025 (Supplementary Fig. 5A). Whereas both PF-07208254 and BT2 demonstrated reductions in hepatic TG content and fasting insulin levels (Fig. 2F, H), PF-07238025 significantly increased liver weight (Supplementary Fig. 5B) and did not alter hepatic TG (Fig. 4G) or fasting insulin levels (Fig. 4H).

PK/PD was assessed to measure blood BCAA/BCKA concentrations timed with the first PF-07238025 dose (Fig. 4I, J), and again after 7 days (Fig. 4K, L). With the first compound dose, a significant reduction in both BCAAs (Fig. 4I, Supplementary Fig. 5C, D) and BCKAs (Fig. 4J, Supplementary Fig. 5E, F) were observed, and BCKAs remained significantly reduced 24 h post dose. However, by day 7, BCAAs were no longer significantly reduced by PF-07238025 (Fig. 4K, Supplementary Fig. 5G, H), and while BCKAs were still acutely reduced, they trended to an increase 24 h post dose (Fig. 4L, Supplementary Fig. 5I, J). After 6 weeks, plasma BCKAs were also reduced at both dose levels (Fig. 4M, N, Supplementary Fig. 5M, N). Similarly, BCAA/BCKA in

gastrocnemius were significantly and dose-dependently reduced 1 h post final compound dose (Supplementary Fig. 5R, S). Interestingly, plasma leucine and valine tended to be increased, and isoleucine was significantly increased above vehicle at the 24 h time point post compound administration (Fig. 4M, Supplementary Fig. 5K, L). Surprisingly, there was also a significant elevation in all BCKAs (Fig. 4N, Supplementary Fig. 5M, N) above vehicle at this time point, suggesting an apparent rebound in BCKA concentrations above baseline. This rebound was apparent despite the observation that circulating PF-07238025 levels showed a consistent daily profile across the study duration (Supplementary Fig. 5O–Q), which suggested that there may be pharmacological differences in PF-07238025 vs. PF-07208254 and BT2.

To understand whether the pharmacology demonstrated with PF-07238025 was common across the series, the related analog PF-07247685 was also subjected to in vivo studies. An 18-day dose ranging study was performed in HFD-fed mice with vehicle or PF-07247685 (Supplementary Fig. 6A), and an oGTT was performed on day 2. While PF-07208254 and BT2, and even PF-07238025, reduced glucose excursion after 2 days (Figs. 2, 4), animals treated with PF-07247685 did not demonstrate an improvement in glucose tolerance at this time point (Supplementary Fig. 6B, C), and even showed a significant worsening of glucose tolerance with the highest dose of PF-07247685 compared with vehicle-treated animals. On day 3, a PK/PD assessment was performed (Supplementary Fig. 6D-P), and AUCs of BCKAs were significantly reduced over 24 h (Supplementary Fig. 6F, J, N). However, by day 17, while AUC of BCKAs for the two higher doses were still significantly reduced, the 24 h time points at the lowest doses demonstrated a significant or nearly significant BCKA rebound above vehicle (Supplementary Fig. 6G, K, O), which occurred in the absence of any changes in plasma drug concentration over the study duration (Supplementary Fig. 6P) and suggested that PF-07247685 and PF-07238025 elicit similar pharmacological responses. In a separate study, HFD-fed animals were dosed with PF-07247685 daily for 9 days, and 1 h post final compound dose, BCAA/BCKA were significantly reduced in gastrocnemius compared with vehicle animals (Supplementary Fig. 6Q–S), suggesting that PF-07247685 acutely lowered tissue BCAA/BCKA.

## Differing pharmacology of thiophene and thiazole BDK inhibitors

To confirm these unexpected results, a two-part study was performed to match BDK inhibitors BT2, PF-07208254, PF-07238025, and PF-07247685 for PD effect to achieve maximal and equivalent reduction of BCAA/BCKA over 24 h. This was necessary to control for differing BDK inhibition potencies and PK profiles. BDK inhibitors PF-07208254, PF-07238025, and PF-07247685 were dosed BID (*bis in die* or twice daily) for 19 days, while BT2 was dosed QD (*quaque die* or once a day) with a vehicle-only second daily dose (Fig. 5A, B). Plasma BCAAs and BCKAs were measured on days 3 (Fig. 5C, F, Supplementary Fig. 7A, D, G, J) and 17 (Fig. 5D, G, Supplementary Fig. 7B, E, H, K) to demonstrate that sustained BCAA and BCKA lowering was achieved for most of the day for all compounds, and the magnitude of effect was similar over the study duration. Compound concentrations were monitored and were consistent between days 3 and 17 (Supplementary Fig. 7M–P). After 19 days, dosing was stopped, and plasma BCAA and BCKA were measured 24, 36, 45, and 68 h post final compound dose (Fig. 5E, H, Supplementary Fig. 7C, F, I, L). The two thiophenes BT2 and PF-07208254 showed a gradual return of BCKAs to baseline over this time course. Interestingly, the two thiazoles PF-07238025 and PF-07247685 demonstrated a profound rebound in BCKA levels upon compound washout, with significantly higher BCKA levels that were nearly 2-fold above the vehicle group, which then returned towards baseline by 68 h post compound dose. These data demonstrated that the two chemical series had very different pharmacological effects on BCKA levels,

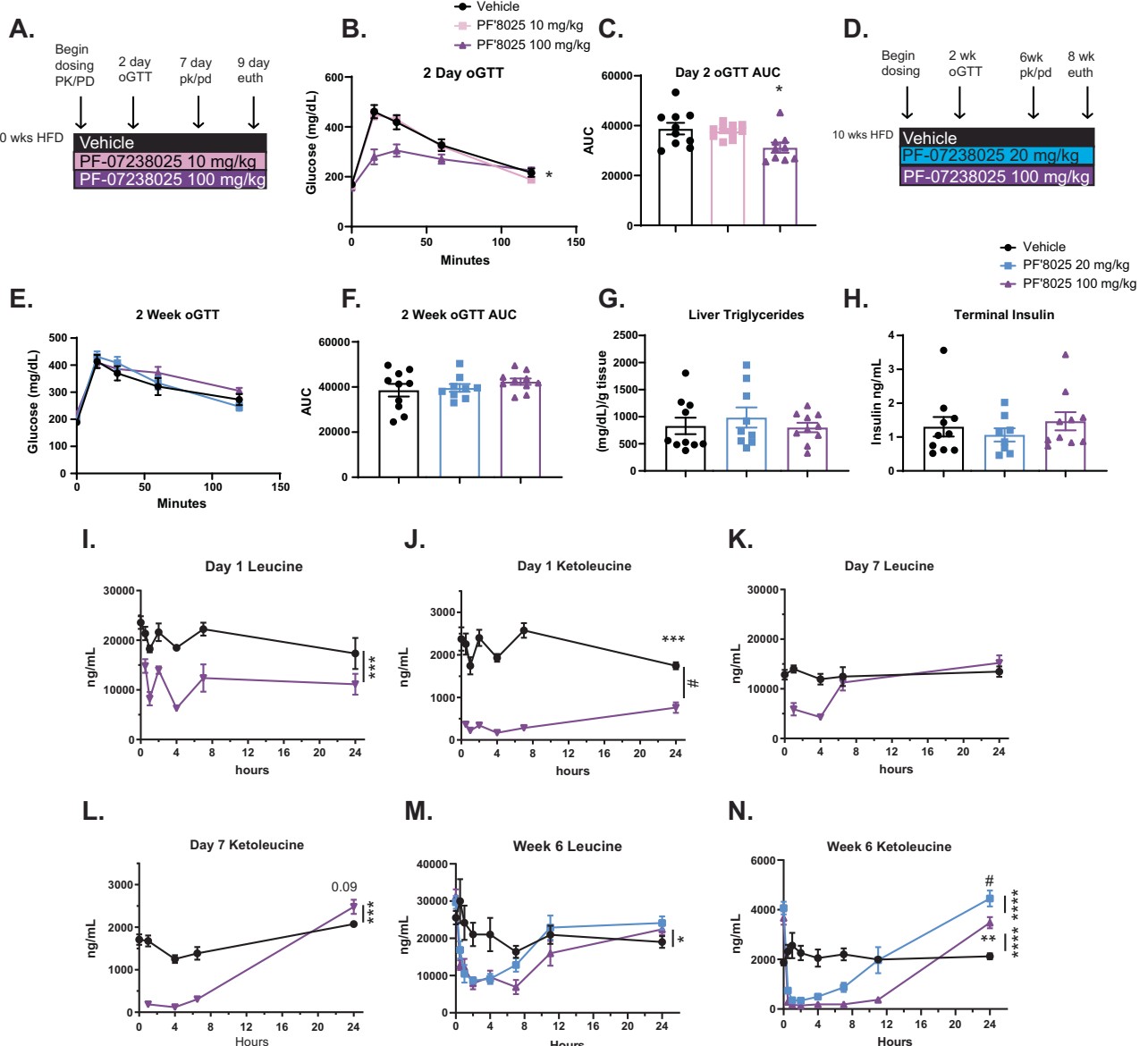

**Fig. 4 | Thiazole compounds do not improve metabolic endpoints after day 2, and BCAA/BCKA rebound occurs over time. A–C** Mice were fed HFD for 10 weeks, at which time animals were randomized into groups, and treated daily with vehicle or PF-07238025 for 9 days ($N$ = 9–10 animals/group; statistics were performed using one-way ANOVA with Tukey HSD test, *$p$ = 0.02). **A** Study design. **B, C** An oGTT was performed at day 2. **B** oGTT. **C** oGTT AUC. **D–N** Mice were fed HFD for 10 weeks, at which time animals were randomized into groups, and treated daily with vehicle or PF-07238025 for 8 weeks. **D** Study design. **E, F** An oGTT was performed at week 2 ($N$ = 9–10 animals/group). **E** oGTT, **F** oGTT AUC. **G, H** Plasma and livers were isolated 1 h post final compound dose. **G** Hepatic triglycerides ($N$ = 9–10 animals/group). **H** Plasma insulin levels ($N$ = 8–10 animals/group). **I–N** PK/ PD assessments were performed on day 1 (**I, J**) ($N$ = 4 animals/group; statistics were performed by Welch's $t$-test), after 7 days (**K, L**) ($N$ = 4 animals/group; statistics were performed by Welch's $t$-test), or 6 weeks (**M, N**) ($N$ = 4–10 animals/group; statistics were performed using one-way ANOVA with Tukey HSD test) of treatment, in which mice were dosed with compound, and timed bleeds were performed to measure drug levels, BCAA, and BCKA levels. **I** Day 1 Leucine (***$p$ = 0.0049), (**J**) Day 1 Ketoleucine (#$p$ = $1.64 \times 10^{-6}$, ***$p$ = 0.003). **K** Day 7 Leucine, (**L**) Day 7 Ketoleucine (***$p$ = 0.001). **M** Week 6 Leucine (*$p$ = 0.019), (**N**) week 6 Ketoleucine (****$p$ < 0.001, ***$p$ = 0.001). Data represent the mean ± SEM. Source data are provided as a Source Data file.

which manifested primarily when compound levels were low. The thiophenes demonstrated prolonged BCKA lowering, whereas the thiazoles demonstrated BCKA rebounding upon compound washout.

To understand the mechanism underlying this unexpected in vivo profile, tissue mRNA levels of BCAA catabolic pathway members were measured after chronic (2–8 weeks) BT2 (thiophene) or PF-07208025 (thiazole) treatment in HFD-fed mice. While both compounds significantly reduced *Bckdk* and other BCAA catabolic pathway gene expression in liver (Supplementary Fig. 8A, B), neither compound had significant effects on *Bckdk* or BCAA catabolic pathway gene expression in striated tissues including skeletal muscle and heart

(Supplementary Fig. 8C, D). The lack of consistent directionality in *Bckdk* expression across tissues did not fully explain the differential regulation of BCKA levels that was observed with the two compound classes.

To test the hypothesis that pBCKDH or BDK protein levels may be altered after chronic treatment with the two BDK inhibitor series, immunoblots were performed from HFD-fed mouse tissues after chronic (2–8 weeks) treatment with thiophenes PF-07208254 and BT2 or thiazoles PF-07238025 and PF-07247685. In heart, pBCKDH/BCKDH was significantly reduced by all four compounds at all doses, demonstrating that all compounds indeed inhibit BDK in vivo. (Fig. 6A–F).

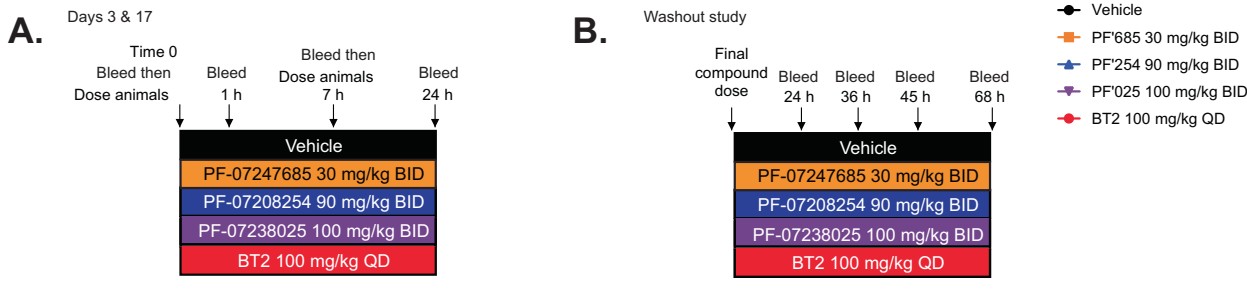

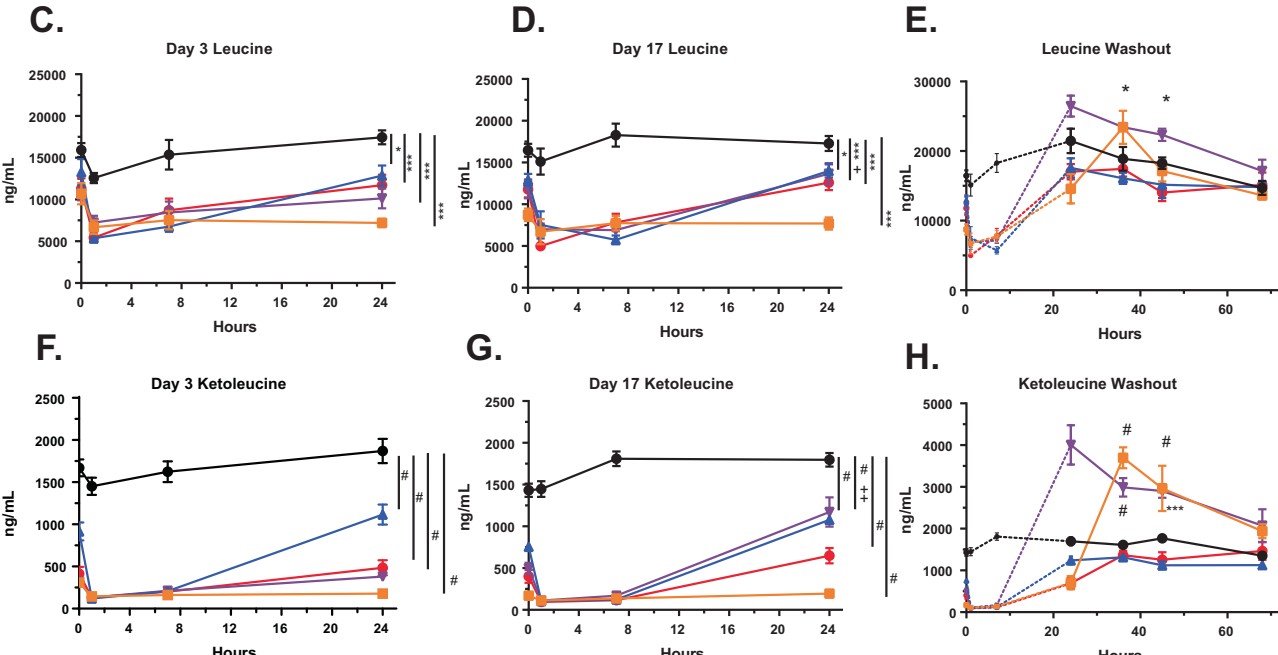

**Fig. 5 | Washout PK/PD demonstrates different pharmacological profiles of thiazole and thiophene BDK inhibitors.** Mice were fed HFD for 10 weeks, at which time animals were randomized into groups, and treated BID or QD as indicated with vehicle, BT2, PF-07208254, PF-07247685, or PF-07238025 at maximal inhibitory doses. **A** Study design. Mice were bled immediately prior to dosing compound, 1 h post compound dose, 4 h post compound dose, 7 h post compound dose (immediately prior to 2nd daily compound dose if BID dosed), and 24 h after first compound dose on day 3 and day 17 ($N = 8$–9 animals/group; a longitudinal mixed effects model with an unstructured covariance was used for statistics for **C**–**D**, **F**–**G**). **B** After the final compound dose on day 19, animals were bled at 24 h, 36 h, 45 h, and 68 h on days 20–21 ($N = 5$ animals/group; a longitudinal mixed effects model with a random intercept for each mouse and an AR(1) covariance structure was fit for statistics for **E**, **H**). **C** Leucine was measured on day 3, (**D**) day 17 (*$p = 0.043$, **$p = 0.01$, ***$p = 0.001$, #$p < 0.0001$, +within group, $p = 0.049$ from day 3), and (**E**) upon compound washout (*$p = 0.047$, 0.013). **F** Ketoleucine was measured on day 3, (**G**) day 17 (#$p < 0.0001$, ++ within group, $p < 0.001$ from day 3), and (**H**) upon compound washout (#$p < 0.0001$, ***$p = 0.001$). Dashed lines in panels **E**, **H** represent the 0–7 h time points from **D** and **G** for visualization purposes. Data represent the mean ± SEM. Source data are provided as a Source Data file.

Consistent with the results in Fig. 1, BDK protein levels were significantly and dose-dependently reduced with PF-07208254 and BT2. Previous reports[4] demonstrated BDK protein reductions upon BT2 treatment, so PF-07208254 behaved similarly to BT2 in this regard (Fig. 6A, B). Surprisingly, in PF-07238025 and PF-07247685-treated animals, BDK protein levels were increased in a dose-dependent manner (Fig. 6C–F). BDK protein levels were also significantly upregulated by thiazoles in gastrocnemius and white adipose tissue (Supplementary Fig. 8E–H). These results potentially explained the differing pharmacology of the two BDK inhibitor classes. While both classes of inhibitors reduced BCKDH phosphorylation and inhibited BDK, there were differing effects on BDK protein levels. These results are consistent with the notion that the BCKA rebound observed with the thiazoles may be due to increased BDK protein levels.

To confirm translation across species, HEK293 cells were treated with thiophenes PF-07208254 or BT2 in a dose response for 48 h to mimic a chronic dosing paradigm, and pBCKDH/BCKDH and BDK levels were measured by immunoblot (Fig. 7A, B). Similar to mice, both PF-07208254 and BT2 significantly reduced pBCKDH as well as BDK protein levels in a dose-responsive manner. Conversely, thiazoles PF-07238025 or PF-07247685 reduced pBCKDH in a dose-dependent manner, and BDK protein levels tended to increase by 50% or greater, similar to mice (Fig. 7C, D). These data suggest that compound series-mediated changes in BDK protein abundance are conserved across species; thiophenes promote BDK degradation, while thiazoles act as BDK stabilizers.

### Mechanism of BDK degradation or stabilization by BDK inhibitors

Because the thiophenes and thiazoles had differing effects on BDK protein levels, we hypothesized that the compounds may alter the BDK interactome. To assess this, BDK was immunoprecipitated (IP'd) from

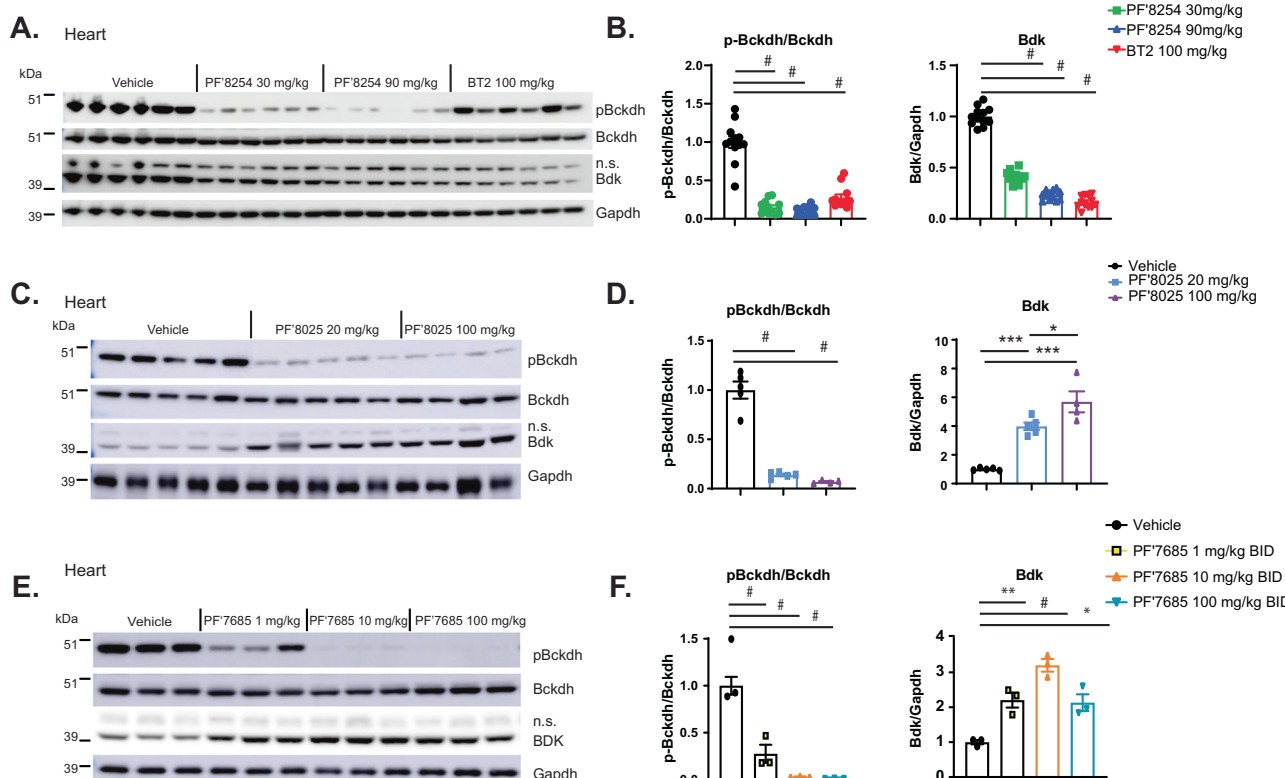

**Fig. 6 | Differing effects of thiazoles and thiophenes on pBCKDH and BDK protein levels in mice. A**, **B** HFD-fed mice were treated with PF-07208254 or BT2 for 8 weeks as described in Fig. 2A. Hearts were isolated and subjected to Western blot for pBckdh, Bckdh, Bdk, and Gapdh. **A** Representative Western blot images, (**B**) Densitometric analyses of pBckdh/Bckdh (left panel) and Bdk/Gapdh (right panel) ($N = 12$ animals/group, $^{\#}p < 0.0001$). **C**, **D** HFD-fed mice were treated with PF-07238025 for 8 weeks as described in Fig. 4D. Hearts were isolated and subjected to Western blot for pBckdh, Bckdh, Bdk, and Gapdh. **C** Representative Western blot images, **D** Densitometric analyses of pBckdh/Bckdh (left panel) and Bdk/Gapdh (right panel) ($N = 4$–5 animals/group, $^{\#}p < 0.0001$, $^{*}p = 0.03$, $^{***}p < 0.001$). **E**, **F** HFD-fed mice were treated with PF-07247685 for 18 days as described in Supplementary Fig. 6A. **E** Representative Western blot images, **F** Densitometric analyses of pBckdh/Bckdh (left panel) and Bdk/Gapdh (right panel). ($N = 3$ animals/group; $^{\#}p < 0.0001$, $^{*}p = 0.011$, $^{**}p = 0.008$). A one-way ANOVA with Tukey HSD test was performed for **B**, **D**, and **F** n.s. non-specific band. Data represent the mean ± SEM. Source data are provided as a Source Data file.

mouse hearts after chronic BT2 treatment, representing the thiophenes, or PF-07247685, representing thiazoles, and untargeted proteomics analysis was performed. Statistical analysis of the top differentially bound proteins in the thiazole series IP represented BDK itself, which was expected as this protein is increased in PF-07247685 treated animals (Fig. 6). Among the other top differentially enriched proteins were Dbt (Bckdhe2), Bckdhb, and Bckdha, all components of the BCKDH complex. Conversely, among the top proteins that bound BDK more after BT2 treatment were proteins involved in protein turnover such as Yme1l1, Smurf1 and Usp2 (Fig. 8A). BDK IP specificity was confirmed by IP and immunoblot (Fig. 8B). First generation BDK inhibitors disrupt the interaction of BDK and the E2 subunit of BCKDH[8]. The IP experiments supported this observation and further suggested that thiophene BDK inhibitors reduced BDK levels by promoting its release from the BCKDH complex leading to its degradation, whereas thiazole inhibitors promoted BDK upregulation by stabilizing it on the BCKDH complex and thus protecting it from degradation.

To further validate this hypothesis, an in vitro protein-protein interaction (PPI) AlphaLISA-based assay was developed to assess the effects of BDK inhibitors on BDK proximity to the full length non-lipoylated BCKDHE2 core subunit (Fig. 8C). Thiophene BDK inhibitors BT2 and PF-07208254 disrupted the BDK and E2 interaction as demonstrated by reduced fluorescence signal with $IC_{50}$ values ± standard error (SE) of $1300 \pm 290$ nM ($n = 26$) and $56 \pm 26$ nM ($n = 5$), respectively, which were similar to their in vitro inhibition potency values (Table 1). These BDK degraders are therefore also destabilizers of the BDK-E2 interaction. In contrast, thiazole BDK inhibitors

PF-07238025 and PF-07247685 increased fluorescence signal with $EC_{50}$ values ± SE of $19 \pm 1.7$ nM ($n = 27$) and $2.2 \pm 0.35$ nM ($n = 5$), respectively, also in line with their in vitro inhibitory potency values (Table 1), suggesting closer BDK-E2 proximity with these compounds, and thus BDK-E2 stabilization. Importantly, inhibitors that reduce fluorescence, indicative of further distance between BDK and E2, also demonstrate lower BDK protein levels, while those that enhance fluorescence demonstrate closer BDK and E2 proximity and BDK accumulation (Fig. 6).

To further understand molecular differences in these chemotypes, molecular dynamics (MD) simulations were employed (Fig. 9A, B, Supplementary Figs. 9, 10). Root-mean-square fluctuations (RMSFs) of all protein Cα atoms with respect to initial structures indicated loop 1 residues Y47–D53 had on average 1.7x larger fluctuations (ranging from 1.2x–2.1x) for destabilizers than for stabilizers, while the same residues in the apo simulations had 1.3x larger average fluctuations (1.1x–1.5x) (Fig. 9A). The RMSF differences in this region are particularly striking when focused on the residues in the BDK lipoyl-binding pocket (Fig. 9B). This pocket binds the E2 lipoyl bearing domain (LBD), which brings E1 into proximity of BDK for subsequent phosphorylation. Thus, binding of destabilizers to BDK may cause the lipoyl-binding pocket/loop 1 (LBP) residues to be more dynamic than in the apo structure, while binding of stabilizers causes these residues to be less dynamic. Visualization of the first principal component (PC1) from essential dynamics analysis of the trajectories demonstrates the importance of this increased flexibility, showing large BDK loop 1 motion in the trajectories of destabilizers, which is reduced in the

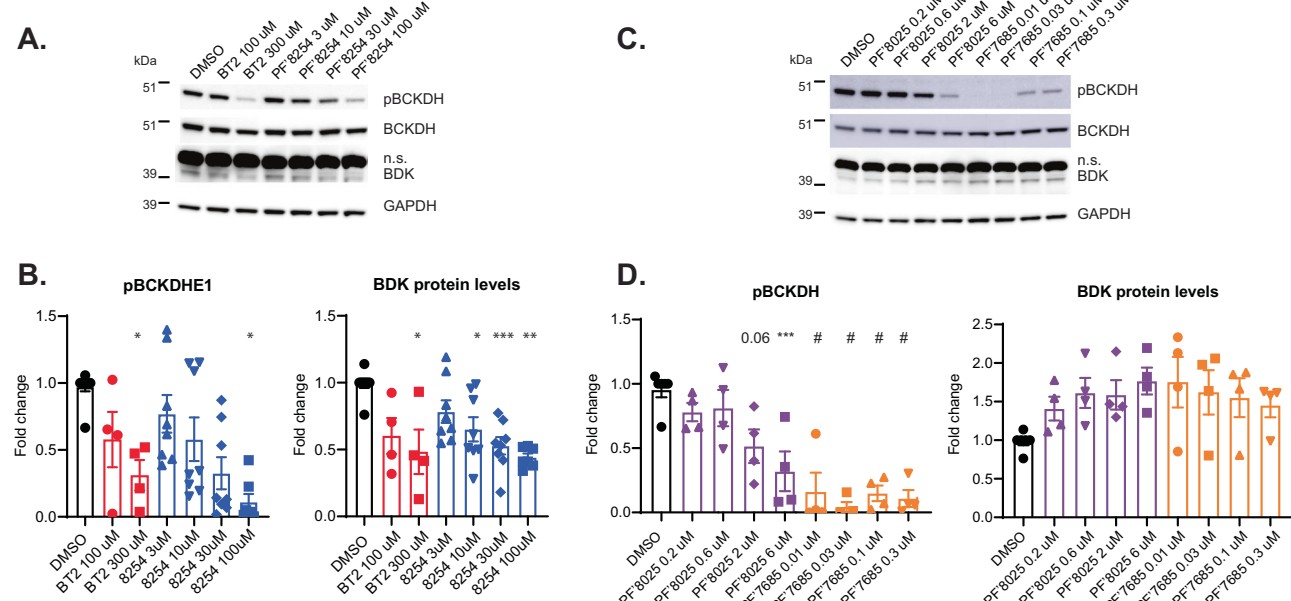

**Fig. 7 | Differing effects of thiazoles and thiophenes on pBCKDH and BDK protein levels in human cells. A–D** Hek293 cells were treated in culture with compounds for 48 h. **A, B** Cells were treated with PF-07208254 or BT2 in a dose response. **A** Representative Western blot images. **B** Densitometric analyses of pBCKDH/BCKDH (left panel) and BDK/GAPDH (right panel) (pBCKDH *p = 0.036, 0.049, BDK *p = 0.014, 0.033 ***p = 0.004, 0.005). **C, D** Cells were treated with PF-07238025 or PF-07247685 in a dose response. **C** Representative Western blot images. **D** Densitometric analyses of pBCKDH/BCKDH (left panel, ***p = 0.002, #p < 0.0001) and BDK/GAPDH (right panel). n.s. non-specific band. (N = 4–8 independent experiments; one-way ANOVA with Tukey HSD test was performed for statistical analyses). Data represent the mean ± SEM. Source data are provided as a Source Data file.

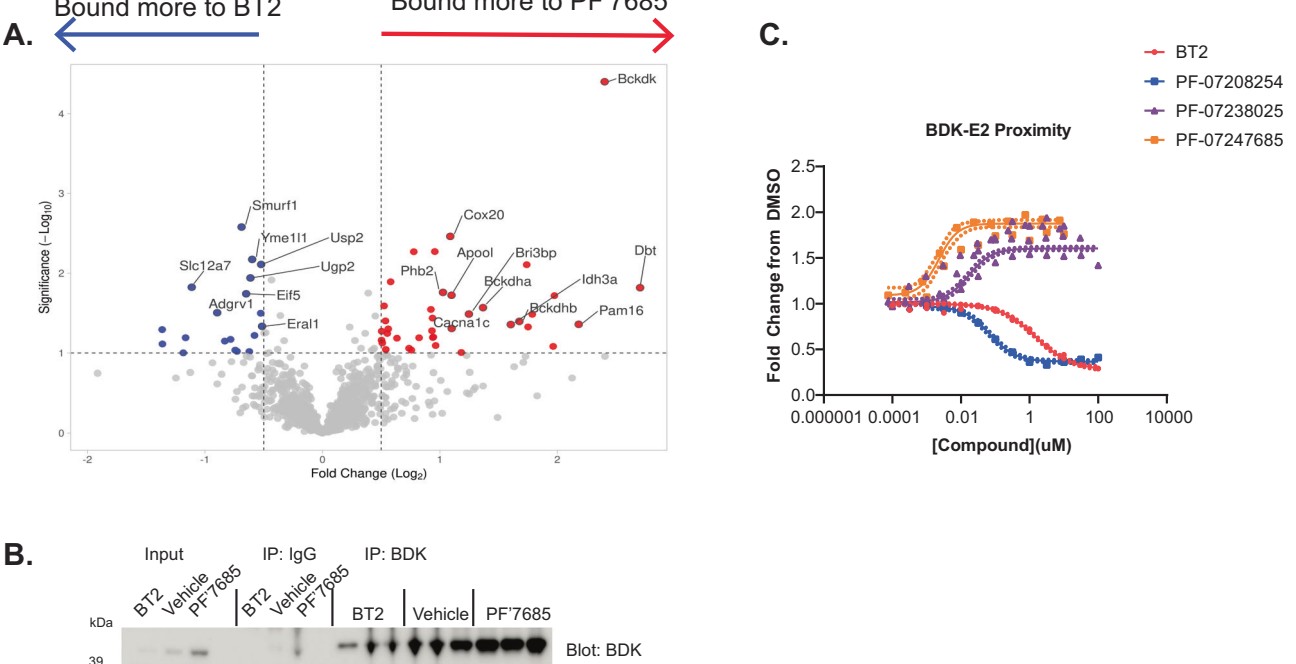

**Fig. 8 | Thiazoles enhance binding of BDK to BCKDH complex while thiophenes reduce binding and promote degradation. A** BDK was immunoprecipitated from hearts of DIO mice that were treated with vehicle, 100 mg/kg BT2 QD or 10 mg/kg PF-07247685 BID for 18 days, and proteomics was performed on the immunoprecipitated proteins. Volcano plot demonstrating the most significantly enriched proteins bound to BDK after BT2 treatment (to the left) versus PF-07247585 treatment (to the right). **B** Representative Western blot of BDK after IgG or BDK immunoprecipitation from mouse hearts (N = 3 animals/group). **C** An AlphaLISA assay was performed with BDK and BCKDHE2. Fluorescence aligned with in vitro potency of BT2, PF-07208254, PF-07238025 and PF-0724785 (n = 5–26 independent experiments). Source data are provided as a Source Data file.

trajectories of the stabilizers (Supplementary Fig. 9B and Supplementary Movies).

MD simulations also suggest differences in the BDK active-site cleft (ASC) when bound to destabilizers or stabilizers. Tso et al.[8] observed a narrowing of the ASC between helices α5 and α8 in the X-ray crystal structure of BDK bound to (S)-CPP causing interference with BDK-E1 association and E1 phosphorylation. This observation was confirmed by tracking the center-of-mass of helices α5 and α8 ($CM_{α5-α8}$) over the course of the MD trajectories in this study. When bound to destabilizers, the average $CM_{α5-α8}$ distance is 21.9 and 21.4 Å, respectively, which is narrower than the apo structure, 22.7 Å (Supplementary Fig. 10A, B). In contrast, the BDK ASC is widened when bound to stabilizers, with an average $CM_{α5-α8}$ distance of 23.4 and 23.2 Å, respectively. Differences in the BDK protein hydrogen bond networks over the simulations provides a rationale for these observations (Supplementary Fig. 10C–F).

Together, results from MD simulations and the PPI assay suggest a potential molecular mechanism for destabilization or stabilization of BDK by thiophenes or thiazoles, respectively. Binding of destabilizers to BDK increases flexibility of the LBP residues, creating conformations that either destabilize or prevent E2 LBD binding. This disrupts the interaction of BDK with the BCKDH complex, exposing BDK to its natural degradation pathway and decreasing protein levels. For transient LBP conformations that can bind the E2 LBD while bound to a destabilizer, the narrowed ASC cannot accommodate E1, preventing phosphorylation, and increased LBP dynamics will eventually lead to destabilization and removal from the complex.

In contrast, binding of stabilizers decreases flexibility of LBP residues, which may stabilize interactions with the E2 LBD, leading to the increased BDK-E2 interaction seen in the PPI assay. The BDK ASC is widened when bound to stabilizers, which should accommodate E1 for phosphorylation. However, PF-07238025 and PF-07247685 protrude into the BDK ASC (Supplementary Fig. 10A, F), which would physically interfere with E1 positioning in the cleft and prevent phosphorylation. BDK is then stabilized on the BCKDH complex, protecting it from degradation and leading to BDK protein accumulation. A model of BDK inhibitors that act as stabilizers or destabilizers is shown in Fig. 9C.

## Discussion

The BCAA catabolic pathway is dysfunctional in a multitude of diseases including metabolic syndrome[2,16,31,32], HF[11,33], NAFLD[34,35], cancer[36], and aging[1,2,10,11,29,36–40]. Further, BDK is an attractive kinase for therapeutic targeting due to its allosteric binding pocket and lack of homology to other protein kinases[4,8]. Thus, improving BCAA catabolism with a BDK inhibitor is a desirable therapeutic strategy. Here, we demonstrate discovery of BDK inhibitors with superior potency to previous inhibitors, and of these, PF-07208254 improves HF and metabolism in mice after chronic administration (Fig. 1). This improvement is associated with reduced BDK protein levels and sustained BCAA/BCKA lowering even at low compound concentrations (Figs. 1, 2, 5, 6, Supplementary Figs. 2, 3, 7, 8). Conversely, thiazoles PF-07238025 and PF-07247685 very potently inhibited BDK activity but did not improve metabolism chronically and were associated with a BCKA rebound (Figs. 4, 5, Supplementary Figs. 5–7) at low compound concentrations due to increased BDK protein content (Figs. 6, 7, Supplementary Fig. 8).

Overall, these results point to the discovery of a class of BDK inhibitors that act as BDK stabilizers and increase BDK protein. In the

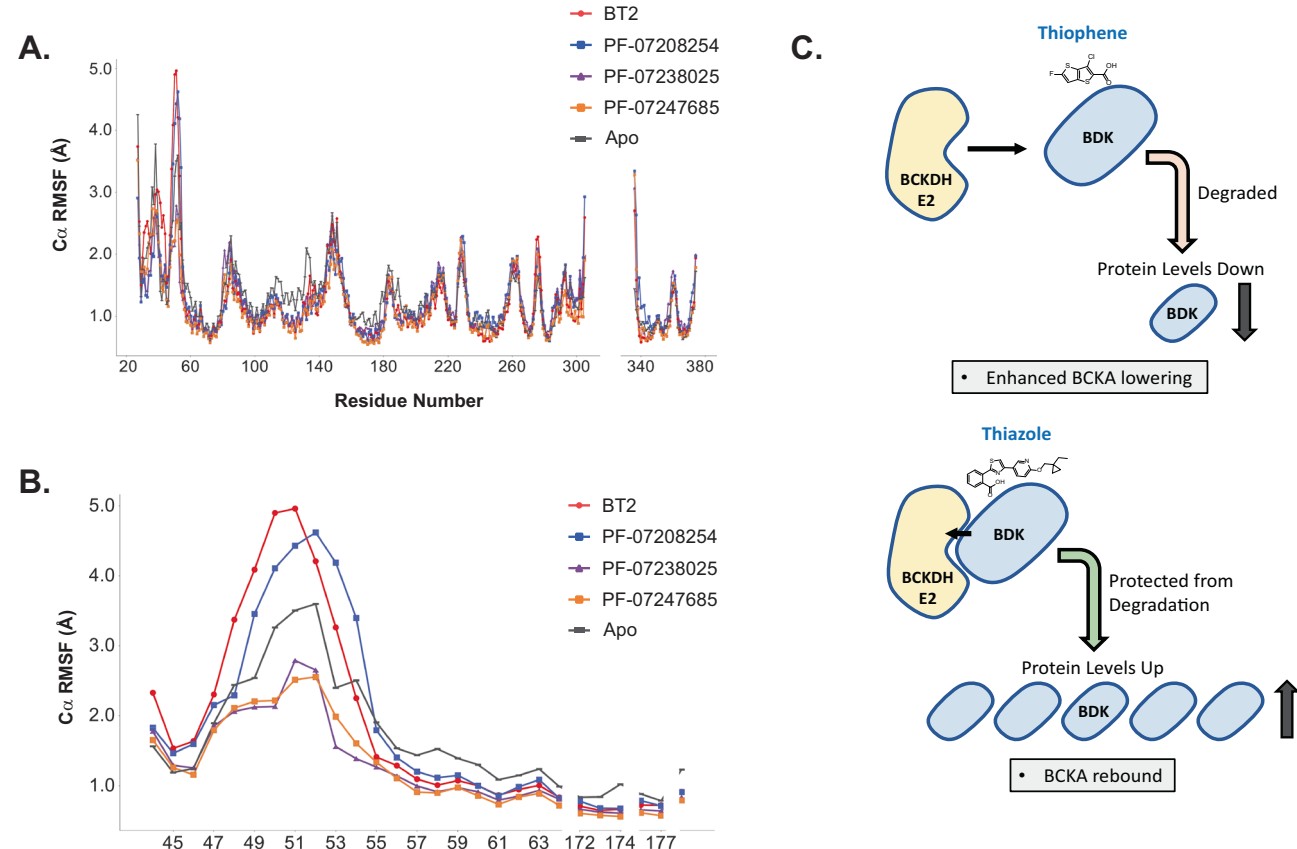

**Fig. 9 | Molecular dynamics simulation shows differing effects on fluctuations of BDK residues. A, B** Root-mean-square fluctuations (RMSFs) based on MD show differences in BDK loop 1 and lipoyl binding pocket residue dynamics when bound to destabilizers or stabilizers. **A** RMSFs of BDK Cα atoms over all residues. **B** RMSFs of loop 1 and lipoyl-binding pocket residues. **C** Model of BDK inhibitor modulation of BDK-BCKDHE2 binding and BDK levels.

case of BDK inhibitors/degraders such as BT2 and PF-07208254, enhanced PD effects are manifested whereby immediate BDK inhibition leads to increased BCAA catabolism. Upon chronic administration, the reduction in BDK protein levels by BDK degraders further enhances BCKA reduction, which is disconnected from PK. For a BDK inhibitor/stabilizer such as PF-07238025 and PF-07247685, when drug is present, immediate BDK inhibition reduces BCKAs; however, upon chronic administration, BDK accumulates and increases BCKAs above baseline once drug levels subside.

The observation that PF-07208254 and BT2 have similar efficacy in preclinical models suggests that they are not efficacious due to an off-target mechanism. Recent studies also demonstrated that BT2 did not improve HF in BDK KO mice[41]. The consistent actions of BDK destabilizers/degraders PF-07208254 and BT2 also suggests that BDK degradation may be beneficial to achieve desired therapeutic effects upon chronic dosing. BDK degradation in HEK293 cells by BT2 and PF-07208254 was slow; a significant reduction in BDK protein levels was only observed after 48 h (Fig. 7). However, BT2 improved insulin sensitivity in as little as 45 min as well as in an oGTT[30], suggesting that BDK inhibition, rather than degradation, must contribute to these effects, as degradation would not have been sufficient within this time frame. Thiazole/stabilizer PF-07238025 also improved glucose tolerance in HFD-fed mice acutely; however, this effect was lost upon chronic dosing. BDK inhibition and BCAA/BCKA reductions alone cannot explain the metabolic efficacy, as thiazole/stabilizer PF-07247685, despite potently inhibiting BDK and reducing BCAA/BCKA, never improved glucose tolerance whether in an acute or chronic setting (Supplementary Fig. 6). BDK and the regulatory phosphatase PPM1k compete for binding on BCKDHE2, and BCKA treatment in vitro can favor PPM1k binding to BCKDHE2, thus facilitating BCKDHE1 dephosphorylation and activation[9]. The thiazoles, while BDK inhibitors, would not favor PPM1k binding since BDK is held on the complex, and thus may outcompete PPM1k. Alternatively, increasing BDK levels and/or the proximity of BDK to the BCKDH complex may promote alternative substrate phosphorylation.

HF efficacy was not assessed with the BDK stabilizers/thiazoles. Based on the metabolic assessments reported here, one may hypothesize that BDK inhibition with a thiazole/stabilizer may not improve HF or could worsen HF. In failing hearts, BCAA catabolic machinery is reportedly reduced, concomitant with increased BCAA and BCKA levels[10,11,13,33], and recent studies have suggested that BCKA transamination can directly promote cardiac hypertrophy in a circadian manner[42,43]. These studies support the notion that increasing BDK levels could further reduce BCAA catabolism, increase BCAA/BCKA levels, and thus drive cardiac pathology. These hypotheses could be addressed in future studies.

Using multiple methods, it was observed that BDK stabilizers promoted interactions between BDK and components of the BCKDH complex. In untargeted IP-proteomics, mice treated with thiazole PF-07247685 showed enriched binding of BCKDH complex components to BDK. One limitation of this assessment is that BDK levels are also increased in the IPs from PF-07247685-treated mice. In vitro AlphaLISA-based assessments also demonstrated increased BDK-BCKDHE2 proximity upon treatment with thiazoles (Fig. 8). Finally, MD simulations demonstrated structural changes upon thiazole binding that may account for this change in association (Fig. 9, Supplementary Figs. 9, 10). The altered proximity and increased binding of BDK to BCKDHE2 and other components of the BCKDH complex likely leads to a protection effect, where BDK becomes protected from its natural degradation cycle and thus, protein accumulates. Mathematical simulations based on the cell culture experiments presented here suggest that BDK half-life may be 100+ hours, which explains how a small change in protein turnover could impact protein levels and why BCKA levels normalize within 3–4 days after BDK inhibitor washout (Fig. 5, Supplementary Fig. 7).

In contrast to BDK stabilizers, using unbiased IP-proteomics from mice treated with BDK destabilizer/degrader BT2, immunoprecipitated BDK bound more to proteins involved in mitochondrial protein homeostasis, protein turnover and the unfolded protein response including Smurf1, Yme1l1, and Usp2 (Fig. 8). These co-immunoprecipitating proteins were enriched even considering that less BDK was pulled down in the IP from BT2-treated versus PF-07247685-treated mice. Using an AlphaLISA-based assay, BDK destabilizers/degraders BT2 and PF-07208254 reduced the proximity of BDK to BCKDHE2, which likely subjected BDK to its natural degradation pathway, thus causing a net reduction in protein levels. Early BDK inhibitors such as (S)-CPP and ketoleucine also demonstrated disruption of the BDK-E2 interaction as shown with isothermal calorimetry[8] and rat liver extract co-IP[44], suggesting that they are also BDK destabilizers/degraders. However, the contribution of BDK degradation vs. inhibition to the efficacy observed in preclinical models was never investigated.

The exact mechanism of BDK degradation by BDK destabilizers is currently unclear. The E3 ligase Ube3b was previously demonstrated to promote BDK degradation in neurons;[45] however, it is uncertain whether this same mechanism is responsible for BT2-mediated BDK degradation. Despite not knowing the exact biochemical machinery, the alteration of BDK co-immunoprecipitating proteins suggests that BDK degraders change the BDK interactome to interact with proteins involved in protein homeostasis rather than the BCKDH complex, which could contribute to the altered physiological profiles observed with these inhibitors.

This work does have several limitations. The in vivo data presented here was performed only in male mice; however, in vitro and cell culture data suggest that the stabilizer and destabilizer pharmacology should not be sex-specific. The metabolic studies performed herein were also limited to HFD-fed mice; db/db mice may be more sensitive to BDK-inhibitor-induced changes as BCAA pathway alterations are apparent in this strain[46], which could be tested in future studies. Further, MD simulations were performed multiple times at 100 nanosecond timescales, and there is some risk that wider conformational changes could be missed by not running MD simulations on the microsecond timescale[47]. Finally, while a subset of BDK inhibitors representing pharmacological chemotypes are presented here, there are other examples of these chemotypes that were not tested in vivo, so it is not certain that every compound in these classes will behave similarly as those presented here.

In summary, we describe a class of BDK inhibitors that stabilizes BDK protein. BDK degraders such as BT2 and PF-07208254 improved metabolism in animal models, whereas BDK stabilizers did not demonstrate this profile upon chronic administration. BDK protein stabilization further led to BCKA rebound upon chronic dose administration at low compound concentrations. The study presented here suggests that BDK degradation may be an important aspect to the pharmacological activity of BDK inhibitors and further defines the desirable profile of an effective therapeutic agent modulating the BCAA metabolic pathway.

## Methods

### Animals

All animal procedures were approved by the Pfizer Institutional Animal Care and Use Committee (IACUC). For HFD-fed mouse studies, 16-week-old male C57BL6/J mice that had been fed Research Diets 12492i for 10 weeks were purchased from Jackson laboratories and were continued on 12492i for the study duration. For transverse aortic constriction studies, 8–10-week-old male C57BL6/N mice were purchased from Charles River. PF-07208254 (1.35 g/kg) or BT2 (0.675 g/kg) were incorporated into Purina 5053 rodent chow by Research Diets. Animals were subjected to 12 h light:dark cycles. Ambient temperature is 72 degrees Fahrenheit with 50% humidity. BT2

was purchased from Enamine (EN300-00845). Compounds were formulated in 0.5% methylcellulose in DI water with 1% (v/v) of Tween 80. Animals were dosed PO QD or BID as indicated. Glucose tolerance tests were performed 1 h post dose after a 16 h fast with 1 g/kg dextrose (Sigma) in water dosed orally using AlphaTrak glucometers and strips on the dog setting. Animals were euthanized with $CO_2$ followed by bilateral pneumothorax, and terminal blood was collected via cardiac puncture.

### Protein harvesting and western blot from tissue
All tissues were snap frozen in liquid nitrogen followed by pulverization with a hammer. 50 mg tissue was lysed in Cell Signaling lysis buffer or Pierce RIPA buffer (for muscle) containing Halt protease and phosphatase inhibitors (Thermo Fisher). Samples were lysed using 2 mL Matrix D Lysis tubes on the MP Fast Prep 24 according to manufacturer's instructions. Samples were spun at 16,260 x $g$ for 10 min, and the supernatant was aliquoted into new tubes. Total protein was quantified using Pierce BCA assay. Samples were diluted to 2 mg/mL and boiled at 95 °C for 5 min with Invitrogen LDS buffer and Reducing Agent, and 10 µL per sample was loaded into NuPage 4–12% Bis-Tris gels. Samples were run on a BioRad system at 135 V for 1 h and 35 min followed by transfer to PVDF membranes. Membranes were blocked in 5% milk in TBS-T, imaged using the Amersham 800 imager, and Imagequant software and Microsoft Excel were used to analyze densitometry. Primary antibodies used are detailed in the Supplementary Methods section.

### Protein-protein interaction AlphaLISA assay
A protein-protein interaction (PPI) assay was set up using the AlphaLISA Surefire technology to assess the effects of compounds on the interaction between biotinylated BDK and the full length non-lipoylated E2 core subunit. Human BCKDH (E1α, residues 46-445) was co-expressed with a pET-DUET construct encoding untagged E1β and E1α with an N-terminal 6xHis, MBP, a TEV protease site, and biotin acceptor peptide. Human DBT (Dihydrolipoamide Branched Chain Transacylase E2), or BCKD-E2, was generated for bacterial expression containing from N- to C-terminus: FLAG peptide, a TEV protease site, and DBT (residues 62-482). Assays were performed in a 10 µL final volume in 384-well low volume plates in buffer containing 20 mM Tris·HCl (pH 7.5), 100 mM KCl, 5 mM $MgCl_2$, 0.5 mM DTT, 0.02% (vol/vol) Tween-20, and 0.1 mg/mL BSA. Compounds were diluted in 100% dimethyl sulfoxide (DMSO) in an 11 dose, half log dilution scheme and spotted in the plates at 100x the final concentration. Compounds were pre-incubated with BDK prior to the addition of the E2 subunit and signal was developed with the addition of 1:1000 CaptSure tagged Ms Mono anti-DBT antibody (Novus), Anti CaptSure acceptor beads and Streptavidin donor beads at 10 µg/µL in Alpha immunoassay buffer (Perkin Elmer) and read in an EnVision® multilabel reader (Perkin Elmer) using Alphascreen settings. The $XC_{50}$ values and maximal response in the E2-BDK assay were generated using ActivityBase software.

### X-Ray structures
X-ray co-crystal structures of human BDK bound to inhibitors have been uploaded to the Protein Structure Database (PDB) as follows: PF-07208254 (PDB 8F5J), **S3** (PBD 8F5S), and PF-07247685 (PDB 8F5F).

### Statistics
A one-way ANOVA was performed for Figs. 2H, 4F–K, Supplementary Fig. 7D. A one-way ANOVA with Tukey HSD test was performed for Figs. 1B–D, I, J, 2I, J, 4C, M (on AUC) 4 N (on AUC and 24 h time point), 6B, 6D, 6 F (compared with vehicle), 7B (compared with vehicle), Supplementary Fig. 2B–K and Supplementary Fig. 3C (ccr2), D–G, J, K, Supplementary Fig. 5B, K–N (on AUC and 24 h time points) R-S, Supplementary Fig. 6B, C (AUC), D-O (AUC), R-S, Supplementary Fig. 7A–L (AUC and 24, 36, 45, 68 h time points), Supplementary Fig. 8C, F, H. A longitudinal mixed effects model with a random intercept for each

mouse and an AR(1) covariance structure was fit for Fig. 5E, H, Supplementary Fig. 2A, Supplementary Fig. 3A and Supplementary Fig. 5A. A longitudinal mixed effects model with a random intercept and an AR(1) covariance structure was fit for each mouse for the glucose AUCs over the course of the study for Fig. 2B–E. A longitudinal mixed effects model with an unstructured covariance was used in Fig. 5C, D, F, G. A Kruskal Wallis test followed by a Dunn test to compare within groups was performed for Fig. 2G. A pairwise Wilcoxon test was performed for Figs. 1E, G, H, 2F and Supplementary Fig. 3B, C (*tnfa, ccl2, cd68, itgam, col1a1, col1a2*). A Welch two-sample *t*-test was performed on AUC for Fig. 4I–L Supplementary Fig. 5C–J, and for the 24 h time point in 2 J, 2 L, Supplementary Fig. 8A, B. All statistical analyses were performed in R 4.0.5

### Additional methods
Off target selectivity panel data and detailed methods for chemical synthesis, mass spectrometry, transverse aortic constriction, echocardiography, HEK293 cell assays, in vitro BDK activity, cellular BDK activity, RNA isolation, liver triglyceride measurement, histopathology, X-ray crystallography, computational methods, and proteomics are available in Supplementary Information.

## Data availability
The mass spectrometry proteomics data have been deposited to the MassIVE repository with the dataset identifier MSV000091341. PDB accession codes 8F5J, 8F5S, and 8F5F can be accessed at: https://doi.org/10.2210/pdb8F5J/pdb, https://doi.org/10.2210/pdb8F5S/pdb, and https://doi.org/10.2210/pdb8F5F/pdb. Source data are provided with this paper.

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

## Acknowledgements

The authors would like to thank Elena Belova, Brian Rago, Natalie Ochocki, Heather Eng, Jie Chen, David Griffith, Ruth Sommese, Timothy Coskran, George Williams, Kelly Tam, Zon Weng Lai and William Esler for helpful discussions or data coordination.

## Author contributions

R.J.R.F., E.B., A.R.R., B.L.K., S.L., R.A.M., B.B.Z., S.K.B., K.O., and K.J.F. conceived of the experiments. R.J.R.F., E.B., A.R.R., B.L., B.L.K., L.A.M.A., L.B., Y.Z., B.B., A.R., P.V.S., J.K., K.O., L.D.H., L.S., L.L., S.B.B., T.C., B.T., F.J.G., J.D., C.P.S., and K.N.Y. performed the experiments. R.J.R.F., E.B., A.R.R., B.L., B.L.K., S.L., M.R.R., L.A.M.A., L.B., Y.Z., B.B., A.R., P.V.S., J.C.S., M.M., B.T., L.X., and A.S.K. analyzed the data. R.J.R.F., B.L.K., S.L., and K.J.F. wrote the manuscript.

## Competing interests

All authors are employees of Pfizer Inc. or were employees at the time the research was conducted. Multiple authors own Pfizer stock. The following authors are inventors on patents or patent applications for BDK inhibitors: R.J.R.F., B.L.K., S.L., M.R.R., L.A.M.A., L.B., Y.Z., S.K.B., K.O., R.A.M., K.J.F.. The other authors have no additional competing interests to declare.
