## [Peer Review File · Nature Communications]

REVIEWER COMMENTS

Reviewer #1 (Remarks to the Author):

Since the BCAA catabolism is related with several diseases, it is important to develop new potent drugs and to clarify mechanisms responsible for the drug actions. This work discovered two types of the very effective drugs against BCKDH kinase (BDK) and demonstrated the differing attributes of the drugs regarding BDK degradation or stabilization. The analyses were conducted very carefully with the high level technics, and the results have a great impact for understanding the regulation of BCAA catabolism.

I have only minor requests to the authors, as follows.

1. Please give comments about classically known BDK inhibitors, such as clofibric acid and thiamine pyrophosphate.
2. Please clarify how much the drugs were contained in the diets for mice in Fig 1.
3. Please specify the meanings of "mpk", "BID", and "QD", because readers other than the pharmacological field are not familiar to these terms.

Reviewer #2 (Remarks to the Author):

Herein Roth Flach and colleagues report the discovery of small molecule BDK inhibitors that have improved potency compared to the widely studied compound BT-2. Two classes of new BDK inhibitors were studied the BT2-like thiophene that inhibited BDK while promoting its dissociation from BCKDH and subsequent degradation and also thiazoles which inhibited BDK but stabilized its interaction with BDK. Although the thiazole compounds were more potent inhibitors of BDK the effect to stabilize BDK protein lead to a rapid rebound in branched-chain ketoacids during drug washout.

While the mechanism of action of the new compounds was well characterized at the molecular level the characterization of physiologic impact of the new compounds is lacking depth and limits the ability to draw conclusions. Major areas of concern are outlined below.

- Authors performed a number of in vivo studies with the new compounds and compared their effects to Bt-2. However, there are no data showing whether the compounds have the same effect on BCKDH systemically as western blotting was performed exclusively in heart. Western blotting for pBCKDH, BCKDH and BDK as well as PPM1K should be conducted in skeletal muscle, liver, BAT, and WAT in addition to heart.

- Authors compare effects of Thiazoles to those of thiophenes and suggest that the impact on physiologic endpoints observed with BT2 are not preserved with Thiazoles. However, whereas the authors studied the thiophenes side-by side with BT2 and performed both TAC and studies with glucose endpoints studies of the Thiazoles were limited to glucose endpoints at 2 days and 2 weeks absent any direct comparison to BT2. Authors should at least complete both studies before reaching the conclusion that the thiazoles are less efficacious. To this end it is noteworthy that an effect on glucose tolerance was observed at 2 days with the thiazoles.

-circulating BCKA and BCAA should be reported for all studies.

- Glucose metabolic effects were studied in high fat fed mice since genetically obese mice display are more prominent induction of BDK and BCAA accumulation these studies should also be performed in this model with each compound.

- If the animals are exposed to the thiazoles throughout the course of the study why would a more rapid washout influence their efficacy for glucose and lipid metabolic endpoints?

- Data in figure 5A are difficult to interpret and should be presented with a data table showing all bound proteins under each condition. A comparison of proteins bound to BDK with vehicle vs each compound would be easier to digest.

- In figure 1, only fractional shortening is identified as significantly different with PF but manuscript reports changes to ejection fraction and heart weight as different language should be adjusted to better reflect the data.

-In figure 1N 2 week OGTT data are shown as AUC absent the glucose excursion these should be included.

- In figure 4 G-H Densitometry does not match western blot. Blot shows pBCKDH effect stronger in BT2 treated samples densitometry shows opposite.

Reviewer #3 (Remarks to the Author):

The manuscript from Filipinski et al disclosing a new thiophene based allosteric BDK inhibitor PF-07208254 and others is well written and thought-out. The work showcases hypothesis driven research with a cutting-edge, full suite of in vitro and in vivo supporting experiments. This work is also well benchmarked against the literature inhibitor standard BT2. The work is not far from publication standard.

It would be good to have a little bit more insight into the medicinal chemistry program behind this story. How big was the HTS screen, hit rate etc. This is likely held for a separate story, but it would be interesting to understand the journey to these compounds. This is quite relevant when PF-07208254 is a morphed replacement of the fused chlorobenzene ring system and PF-07238025/PF-07247685 both have carboxylic acids and related motifs. The authors have partly rationalised this in the docking/modelling but it would be useful to have a better insight here if possible.

Extended Data Figure 8 - 1 micro second simulations would be more relevant and could provide additional insights, the authors could also run these and put them into the supporting information if nothing further is found.

Line 442-447 is an effective summary of some limitations of this current study. It is never possible to know everything, I am not sure how relevant line 444-447 is, but this is to the authors discretion to keep this or not.

The chemistry section has some issues that need to be addressed -

Characterisation of all the products is quite light. I believe the compounds are as reported, but I would expect to see ¹H, ¹³C, HRMS and an LC/HPLC trace to demonstrate purity for each compound. These are production amounts of material reported, so it should be straightforward to collect this data if not already in house.

Supp Line 364-365 '(Additional material, 19.8 g of S15 was obtained in a slightly impure state.)' this is useful for completeness but is not a standard reporting format. This should be addressed.

The supporting information showing all the spectra should be included in the re-submission.

Reviewer #4 (Remarks to the Author):

The manuscript by Roth Flach et al "Discovery of small molecule branched-1 chain ketoacid dehydrogenase kinase (BDK) inhibitors with opposing effects on BDK protein levels" reports data and the discovery of two BDK inhibitor series with different mode of action. Thiazole based inhibitors were shown to promote the BDK-E2 interaction and increased BDK protein levels, thus explaining the BCKA rebound observed and lack of metabolic efficacy. In contrast, the developed allosteric thiophene BDK inhibitor PF-07208254, which had improved potency compared to the known compound BT2, promoted BDK degradation resulting in sustained BCKA lowering in animals. Both compound series were tested in animal models demonstrating efficacy of PF-07208254 and BT2 in HF models including transverse aortic constriction (TAC) and improved glycaemia.

Mechanistically, the authors showed that after BT2 treatment, BDK bound more to proteins involved in protein degradation such as Yme111, Smurf1 and Usp2, thus thiophene BDK inhibitors rewire the interaction network cause degradation of the atypical kinase BDK. Crystal structures were determined of both inhibitors types and together with MM simulation suggested a model which points to the importance of the C α 5- α 8 distance which is narrower in destabilizing compounds (21.9 and 21.4 Å) compared to stabilizers (23.4 and 23.2 Å). This is a highly interesting report of the development and the characterization of two small molecule modulators of BDK that have different modes of actions resulting in diverse metabolomic and pharmacological consequences. I therefore strongly support publication of this study. I have however a number of concerns that the authors should address before publication of these data:

The western blot shown in 1F shows two bands. The upper band seems to be induced during TAC and it is not affected by either PF`8254 nor BT2. Are these bands due to different phosphorylation states of BDK?

The resolution of the three crystal structures is quite low and there seems to be some issues with the diffraction data quality. The B-values for protein and ligands are >100 suggesting that there was not much supporting density building the models. It would be good to split the B-values for the protein atoms in main and side chains to judge if this problem was only due to side chain flexibility. Electron density (OMIT) maps should be shown for the ligands. The high B-values may also due to a refinement and/or scaling problem. How high were the Wilson B-values?

Reviewer #1 (Remarks to the Author):

Since the BCAA catabolism is related with several diseases, it is important to develop new potent drugs and to clarify mechanisms responsible for the drug actions. This work discovered two types of the very effective drugs against BCKDH kinase (BDK) and demonstrated the differing attributes of the drugs regarding BDK degradation or stabilization. The analyses were conducted very carefully with the high level technics, and the results have a great impact for understanding the regulation of BCAA catabolism.

We thank the reviewer for this positive feedback on our study.

I have only minor requests to the authors, as follows.

1. Please give comments about classically known BDK inhibitors, such as clofibric acid and thiamine pyrophosphate.

Text has been added to Line 84 to include clofibric acid, whose binding site has not been determined but, based on structure similarity, is likely the same allosteric pocket, and for thiamine pyrophosphate (TPP, line 85). TPP has not been shown to directly bind to BDK and since its literature data used only partially purified reagents and knowing its involvement as a cofactor in the substrate BCKDH's activity, this should be cautiously referred to as a BDK inhibitor. References for both compounds have also been added as reference # 23 and 24.

2. Please clarify how much the drugs were contained in the diets for mice in Fig 1.

The amount of compound used in the diet was included in the supplemental methods section, as the methods section was quite extensive for this manuscript. Lines 608-610 in the supplemental information stated: "PF-07208254 (1.35 g/kg) or BT2 (0.675 g/kg) were incorporated into Purina 5053 rodent chow by Research Diets". We have now moved this information from the methods supplement to the main methods section (page 20 lines 482-484) so that the information is easier to find.

3. Please specify the meanings of “mpk”, “BID”, and “QD”, because readers other than the pharmacological field are not familiar to these terms.

We have aligned all instances of mpk to mg/kg throughout the manuscript. We have now defined BID and QD at first use on page 10 lines 255-256.

Reviewer #2 (Remarks to the Author):

Herein Roth Flach and colleagues report the discovery of small molecule BDK inhibitors that have improved potency compared to the widely studied compound BT-2. Two classes of new BDK inhibitors were studied the BT2-like thiophene that inhibited BDK while promoting its dissociation from BCKDH and subsequent degradation and also thiazoles which inhibited BDK but stabilized its interaction with BDK. Although the thiazole compounds were more potent inhibitors of BDK the effect to stabilize BDK protein lead to a rapid rebound in branched-chain ketoacids during drug washout.

While the mechanism of action of the new compounds was well characterized at the molecular level the characterization of physiologic impact of the new compounds is lacking depth and limits the ability to draw conclusions. Major areas of concern are outlined below.

- Authors performed a number of in vivo studies with the new compounds and compared their effects to Bt-2. However, there are no data showing whether the compounds have the same effect on BCKDH systemically as western blotting was performed exclusively in heart. Western blotting for pBCKDH, BCKDH and BDK as well as PPM1K should be conducted in skeletal muscle, liver, BAT, and WAT in addition to heart.

We have now included Western blots from skeletal muscle and white adipose tissue. We regret that we did not have liver or BAT collected from animal studies. The results from the gastrocnemius and WAT were qualitatively similar to what was observed in heart tissue. BDK protein is significantly increased with thiazole treatment in both WAT and skeletal muscle. BDK protein is low in both of these tissues, therefore observing further reduction with BT2 or PF-07208254 treatment is challenging as it is at the lower limit of detection by Western blot. In gastrocnemius, reduced pBCKDH/BCKDH is observed with PF-07208254 and the two thiazole compounds but was not observed with BT2 (similar to what was reported in Bollinger et al, Molecular Metabolism 2022 with BT2). Significant changes in the pBCKDH/BCKDH ratio were not observed in white adipose tissue with any compound, similar to what we have previously reported with BT2 (Bollinger et al., 2022). Interestingly, despite the lack of pBCKDH change in WAT, there was still a dramatic BDK upregulation observed in this tissue. The inclusion of these three distinct tissues (heart, skeletal muscle, white adipose tissue) we feel is sufficient evidence that the thiazoles are having BDK stabilization effects systemically. These new data including skeletal muscle, WAT, blots for pBCKDH, BCKDH and BDK are included in Extended data Figure 8 E-H and described in the text page 12 lines 291-292.

We purchased 4 commercially available antibodies for Ppm1k from different vendors (ProteinTech, Abcam and Invitrogen) and have tested them in multiple tissues (heart, WAT and gastroc), but we have been unable to obtain reliable Western blots with any of these antibodies. With each of the antibodies we tested, there were many bands in every tissue, and it is not clear which band is the correct band. Therefore, we regret that we are unable to include Ppm1k blots with the revision.

- Authors compare effects of Thiazoles to those of thiophenes and suggest that the impact on physiologic endpoints observed with BT2 are not preserved with Thiazoles. However, whereas the authors studied the thiophenes side-by side with BT2 and performed both TAC and studies with glucose endpoints studies of the Thiazoles were limited to glucose endpoints at 2 days and 2 weeks absent any direct comparison to BT2. Authors should at least complete both studies before reaching the conclusion that the thiazoles are less efficacious. To this end it is noteworthy that an effect on glucose tolerance was observed at 2 days with the thiazoles.

We acknowledge that BT2 was not included side by side with the thiazole compounds in the studies reported here. BT2 has been published several times including by our own group (Bollinger et al, Molecular Metabolism December 2022), and the effects on metabolism in diet induced obese mice is well established. In studies included in Figure 1 here, as well as the previously reported manuscript (Bollinger et al Molecular Metabolism 2022), BT2 (and PF-07208254) improved glycemia, liver fat and hyperinsulinemia. In the studies reported here in figure 4 with the thiazole PF-07238025, there was no improvement in glycemia, liver fat or hyperinsulinemia in HFD-fed mice after 8 weeks of treatment.

We did not test the effects of glycemia or other metabolic endpoints in HFD-fed mice with PF-07247685 in studies beyond 2 weeks in duration so we cannot comment on whether some effects could be apparent beyond this time point, but as the metabolic effects of BT2 occur very rapidly (~45 min) in rodents (Bollinger et al, Molecular Metabolism 2022), it is unlikely that PF-07247685 would have effects at later timepoints that did not manifest early on. Further, we have acknowledged that we did not test either thiazole molecule in a heart failure study (which is a significant body of work that could be a stand-alone manuscript) in the discussion section page 17 line 416-423. Nonetheless, we have modified the language as it relates to thiazoles and efficacy on page 1 line 52, page 4 line 95, 98-99, page 19 line 458, 470.

-circulating BCKA and BCAA should be reported for all studies.

Circulating BCAA/BCKA for TAC study reported in Figure 1A is included as Figure 11-J & Extended Data 2H-K. BCAA/BCKA for the study reported in Figure 1K was reported in Figure 1R-S and Extended Data 3D-G. BCAA/BCKA for the study reported in 2C&F are reported as Figure 2K-P and Extended Data Figure 5. We have now added the other 4 BCAA/BCKA for days 1 and 7 at the reviewer's request as part of Extended data figure 5C-N manuscript text page 9 lines 215-230. Leucine/Ketoleucine were reported for the study included in Figure 3 as 3C-H. We have now added Isoleucine/Ketoleucine and Valine/Ketovaline at the reviewer's request as Extended data 7 A-L (manuscript text pages 10-11 lines 251-272). Ketoleucine was reported for the study included in Extended data 6 as Extended data 6D-E, we have now added

Leucine, Isoleucine/Ketoisoleucine, Valine/Ketovaline at the reviewer's request as Extended data 6D-P (manuscript text pages 9-10 lines 231-249).

- Glucose metabolic effects were studied in high fat fed mice since genetically obese mice display are more prominent induction of BDK and BCAA accumulation these studies should also be performed in this model with each compound.

We appreciate that there are metabolic differences between db/db and HFD-fed mice including changes in BCAA metabolism as reported in Neinast et al, Cell Metabolism, 2018. However, we do not feel that the addition of db/db animal studies would dramatically change the interpretation of the work presented here in this manuscript. In addition to the work shown here in HFD-fed mice to demonstrate changes in BDK protein levels and BCAA/BCKA levels, we have demonstrated that the thiazole and thiophene compounds have differing effects on BDK protein levels in a human cell culture system (Figure 4), suggesting that this observation translates to different species and is not just phenomenology of HFD-fed mice. Also, in vitro experiments show differences in protein-protein interaction between the chemical series, which would again not be species- or model- specific (Figure 5). It is possible that a db/db model shows differing metabolic results than a HFD-model, which could be due to differences in dietary protein content or altered expression/phosphorylation state of BCKDH. However, we feel that the orthogonal systems already included provide confidence that the main conclusions of the manuscript would not be compromised. We regret that we are not able to provide this study due to resource limitations within Pfizer. We have added the use of only HFD-fed animals as a limitation to the paper in the discussion section to highlight this limitation for the reader page 19 lines 461-464.

- If the animals are exposed to the thiazoles throughout the course of the study why would a more rapid washout influence their efficacy for glucose and lipid metabolic endpoints?

We hypothesize that BDK is protected on the BCKDH complex by the thiazoles, leading to the increased protein levels observed and supported by data from figure 5. Because BDK and PPM1k compete for binding on the BCKDH complex, it is possible that thiazoles may inhibit BDK part of the day and prevent full activation of BCKDH by PPM1k by blocking its access to the complex. Once thiazoles have washed out BDK would again phosphorylate BCKDH and turn it "off". This would explain the BCKA rebound and could lead to reduced efficacy of a BDK inhibitor over time. We have discussed this hypothesis in the discussion section, page 17, lines 410-415.

- Data in figure 5A are difficult to interpret and should be presented with a data table showing all bound proteins under each condition. A comparison of proteins bound to BDK with vehicle vs each compound would be easier to digest.

We agree with the reviewer that all bound proteins under each condition should be reported. Therefore, we have uploaded the entire proteomics dataset of interacting proteins as a supplemental dataset to this manuscript titled "BDK interacting proteins". Additionally, the mass spectrometry proteomics data have now been deposited to the MassIVE repository (<https://massive.ucsd.edu/>) with the dataset identifier MSV000091341. Text has been added to the end of the manuscript to detail this. During the review process, the dataset including

supplemental tables can be assessed using username: MSV000091341_reviewer and password: pfizerbdk.

- In figure 1, only fractional shortening is identified as significantly different with PF but manuscript reports changes to ejection fraction and heart weight as different language should be adjusted to better reflect the data.

We have adjusted the language to read “both PF-07208254 and BT2 improved FS% with a trend towards improvement in EF%” on page 5 line 134-135 and “However, heart weight trended towards reduction by PF-07208254 and was significantly reduced by BT2 treatment, while lung weights trended to reduction with both compounds” on page 6 lines 140-142.

-In figure 1N 2 week OGTT data are shown as AUC absent the glucose excursion these should be included.

We have now included the glucose excursion curve as 1N.

- In figure 4 G-H Densitometry does not match western blot. Blot shows pBCKDH effect stronger in BT2 treated samples densitometry shows opposite.

The densitometry in 4H is an average of 4 experiments, and it is notable that the variation in BT2 treated cells was quite high. We have chosen a different representative blot to include in the image for 4G.

Reviewer #3 (Remarks to the Author):

The manuscript from Filipinski et al disclosing a new thiophene based allosteric BDK inhibitor PF-07208254 and others is well written and thought-out. The work showcases hypothesis driven research with a cutting-edge, full suite of in vitro and in vivo supporting experiments. This work is also well benchmarked against the literature inhibitor standard BT2. The work is not far from publication standard.

It would be good to have a little bit more insight into the medicinal chemistry program behind this story. How big was the HTS screen, hit rate etc. This is likely held for a separate story, but it would be interesting to understand the journey to these compounds. This is quite relevant when PF-07208254 is a morphed replacement of the fused chlorobenzene ring system and PF-07238025/PF-07247685 both have carboxylic acids and related motifs. The authors have partly rationalised this in the docking/modelling but it would be useful to have a better insight here if possible.

*Additional text was added to Line 105-106 to detail the number of compounds screened which led to the hits. Only ~12000 compounds were screened which would not fall into our definition of High Throughput, and instead is targeted screening. 3.6% of those compounds showed single point activity >30%. 49% of those showed an IC50 < 40 uM. This hit rate data is included here but not in the text due to the subjective nature of drawing activity thresholds to determine “hits” and since the screening effort is not the focus of this manuscript. Additional text regarding non-polar side chain contacts was added to Line 123 to rationalize the superior potency of PF-07208254 relative to benzothiophene BT2. Text was added to Line 181 to clarify where the **S3***

thiazole hit originated. Modifying **S3** to arrive at the two key compounds PF-07238025/PF-07247685 involved building into the lipophilic end of the allosteric pocket to improve potency, as mentioned in the text, and the addition of polar atoms which helped to reduce clearance and favor the active conformation.

Extended Data Figure 8 - 1 micro second simulations would be more relevant and could provide additional insights, the authors could also run these and put them into the supporting information if nothing further is found.

It has been demonstrated in the literature that the results from one long molecular dynamics (MD) simulation are often not reproducible and that conclusions drawn based on the analysis of multiple shorter MD replicas starting from the same equilibrated structure with different initial randomized velocities are more reliable and reproducible (Knapp et al. JCTC, 2018, 14, 6127; Vassaux et al. JCTC, 2021, 17, 5187). In addition, it has been shown that the analysis of multiple shorter MD replicas is consistent with experimental results from NMR studies, suggesting that multiple replicates capture experimentally determined protein dynamics (Poongavanam et al. J. Med. Chem. 2022, 65, 13029; Ball et al., Biophys. J., 2019, 116, 1432). With this information in mind, the decision was made to run six independent MD simulations of 100 ns each for all five systems starting with the same equilibrated structure and using new initial randomized velocities, as opposed to running one longer simulation. Running an additional 5 μ s of MD simulations would not be expected to change the results and may potentially even lead to misleading results based on the literature cited above. The hypotheses generated from the MD simulations are based on state-of-the-art computations. Currently, the SI MD methods section (SI Lines 603-608) details this thinking and references the Knapp paper. The other three references above have since been added to the section to further strengthen our logic (SI references 20-22).

Line 442-447 is an effective summary of some limitations of this current study. It is never possible to know everything, I am not sure how relevant line 444-447 is, but this is to the authors discretion to keep this or not.

We thank the reviewer for acknowledging that no study can be 100% comprehensive. While we also acknowledge this, we will keep these discussion points in the manuscript.

The chemistry section has some issues that need to be addressed -

Characterisation of all the products is quite light. I believe the compounds are as reported, but I would expect to see ¹H, ¹³C, HRMS and an LC/HPLC trace to demonstrate purity for each compound. These are production amounts of material reported, so it should be straightforward to collect this data if not already in house.

More thorough characterization data has been added to the SI including ¹³C NMR, ¹⁹F NMR, HRMS, and HPLC. Due to the need to generate additional characterization data with existing samples, the included route to PF-07208254 was somewhat altered. This is shown with an updated synthesis scheme and experimental procedures for Steps 1-4. Furthermore, the generation of additional data to provide PDFs of spectra and chromatograms, in some cases, included retaking ¹H NMR and MS. There are some cases where the chemical shift or mass ion values have changed slightly (e.g., 7.40 vs. 7.41 ppm) from the original submission for this reason. For certain unstable intermediates such as the acetals, limited data is provided since

the length of the characterization method (e.g., ¹³C NMR) is longer than the solution stability of the intermediate. All final compounds, however, include the full suite of data.

Supp Line 364-365 '(Additional material, 19.8 g of S15 was obtained in a slightly impure state.)' this is useful for completeness but is not a standard reporting format. This should be addressed.

This line has been deleted. It did not add meaningfully to that experimental procedure. The current text is accurate and clearer.

The supporting information showing all the spectra should be included in the re-submission.

Images of all spectra and chromatograms have been added to the SI as a separate file.

Reviewer #4 (Remarks to the Author):

The manuscript by Roth Flach et al "Discovery of small molecule branched-1 chain ketoacid dehydrogenase kinase (BDK) inhibitors with opposing effects on BDK protein levels" reports data and the discovery of two BDK inhibitor series with different mode of action. Thiazole based inhibitors were shown to promoted the BDK-E2 interaction and increased BDK protein levels, thus explaining the BCKA rebound observed and lack of metabolic efficacy. In contrast, the developed allosteric thiophene BDK inhibitor PF-07208254, which had improved potency compared to the know compound BT2, promoted BDK degradation resulting in sustained BCKA lowering in animals. Both compound series were tested in animal models demonstrating efficacy of PF-07208254 and BT2 in HF models including transverse aortic constriction (TAC) and improved glycaemia.

Mechanistically, the authors showed that after BT2 treatment, BDK bound more to proteins involved in protein degradation such as Yme1i1, Smurf1 and Usp2, thus thiophene BDK inhibitors rewire the interaction network cause degradation of the atypical kinase BDK. Crystal structures were determined of both inhibitors types and together with MM simulation suggested a model which points to the importance of the C α 5- α 8 distance which is narrower in destabilizing compounds (21.9 and 21.4 Å) compared to stabilizers (23.4 and 23.2 Å). This is a highly interesting report of the development and the characterization of two small molecule modulators of BDK that have different modes of actions resulting in diverse metabolomic and pharmacological consequences. I therefore strongly support publication of this study. I have however a number of concerns that the authors should address before publication of these data:

The western blot shown in 1F shows two bands. The upper band seems to be induced during TAC and it is not affected by either PF`8254 nor BT2. Are these bands due to different phosphorylation states of BDK?

There is a band that runs above the BDK band, which is prominent in some tissues and in cells (see Fig 4 for additional Western blots of BDK). It does appear that the band is a bit darker in the TAC animals, but we do not think that the upper band is BDK due to QC checks we have performed internally. We ran a Coomassie gel and cut out pieces of the gel corresponding to the "upper band" and "lower band" and ran mass spectrometry on the pieces. BDK peptides were only obtained in the "lower band" and not the upper band. In mice treated with the thiazole PF-`8025, the abundance of these peptides increased in the "lower band", which correlated with the Western blot analysis. See below for quantitation of BDK protein in the lower band. We do not

know what the upper band is on the gel, but we do not think it is BDK and therefore should not factor into the data interpretation.

Figure: Proteomics of “bottom band” from protein lysates collected from the study performed in Figure 2. Left 2 samples are thiazole-treated, right 2 samples are vehicle-treated.

The resolution of the three crystal structures is quite low and there seems to be some issues with the diffraction data quality. The B-values for protein and ligands are >100 suggesting that there was not much supporting density building the models. It would be good to split the B-values for the protein atoms in main and side chains to judge if this problem was only due to side chain flexibility. Electron density (OMIT) maps should be shown for the ligands. The high B-values may also due to a refinement and/or scaling problem. How high were the Wilson B-values?

*The X-ray table, Table S4, has been updated. For compound **S3** and PF-07247685, there are two protein complexes in the crystal asymmetric unit. Chain A is more ordered than chain B with lower temperature factors, but both have good electron density for ligands. During model building and refinements chain B was tightly restrained to chain A. The updated table contains overall average B factors listed separately for chain A and B of proteins, and bound ligands. The differences between the main chain and side chains are not large. The overall Wilson plot B factors of data sets used are also included. From the table the structures are clearly well refined. The OMIT maps are added as new data in Extended data Figure 4A-C.*

We again thank the editor for the chance to revise our manuscript, and we thank the reviewers for their time and for excellent and insightful suggestions. We feel that with these additional data and clarifications that our manuscript is much improved and is now hopefully acceptable for publication in Nature Communications.

Best Regards,

Rachel J. Roth Flach, PhD
Associate Research Fellow
Internal Medicine Research Unit
Pfizer, Inc
Cambridge, MA 02139

Kevin J. Filipski
Associate Research Fellow
Medicine Design
Pfizer, Inc
Cambridge, MA 02139

REVIEWER COMMENTS

Reviewer #2 (Remarks to the Author):

The authors have adequately addressed my concerns.

Reviewer #3 (Remarks to the Author):

The authors have significantly improved the manuscript and fully addressed most points raised.

However, one point was only partially addressed, relating to the MD simulations within the paper. I was not suggesting running 1 long 5 microsecond MD simulation, rather 5×1 microsecond. I agree with the authors that running 6×100 nS is better than 1×600 nS. I agree with the response in part, but there is a risk that wider conformational changes could be missed as evidenced by Kern et al. *Nature*, 450, p 913–916 (2007) - <https://www.nature.com/articles/nature06407>. This is particularly relevant as while a number of observations are made based on the results of these MD's (which are supported). The point being missed, is that with a small amount of additional outlay potentially valuable information could be gleaned; from a system where the authors already have extensive understanding.

Reviewer #4 (Remarks to the Author):

The authors addressed all issues I raised in my review. I have therefore no concerns recommending the current version of the manuscript for publication.

A response to the concerns raised by reviewer #3 is added below in italics.

Reviewer #3 (Remarks to the Author):

The authors have significantly improved the manuscript and fully addressed most points raised.

However, one point was only partially addressed, relating to the MD simulations within the paper. I was not suggesting running 1 long 5 microsecond MD simulation, rather 5 x 1 microsecond. I agree with the authors that running 6 x 100 nS is better than 1 x 600 nS. I agree with the response in part, but there is a risk that wider conformational changes could be missed as evidenced by Kern et al. Nature, 450, p 913–916 (2007) - <https://www.nature.com/articles/nature06407>. This is particularly relevant as while a number of observations are made based on the results of these MD's (which are supported). The point being missed, is that with a small amount of additional outlay potentially valuable information could be gleaned; from a system where the authors already have extensive understanding.

We recognize the discerning point raised by Reviewer 3 that there is a risk of missing wider conformational changes in these systems by not running molecular dynamics (MD) simulations on the microsecond timescale. The timescale in which we ran our MD simulations was 5x 100 ns each, which was a conscious choice based on our internal computational capabilities as well as evidence from recent literature that demonstrates that analysis of multiple shorter MD replicas is consistent with experimental results from NMR studies, suggesting that multiple replicas capture experimentally determined protein dynamics.

We of course wanted to run these MD simulations for as long as possible; however, running microsecond timescale MD simulations for the five systems in the manuscript is not a small amount of additional outlay for us. Based on our current computational capabilities, we are able to run ~100 ns of production MD / day for these systems, corresponding to 10 full days to achieve 1 microsecond of production simulation time for one system. Running 5 x 1 microsecond for each system, as suggested by Reviewer 3, would take 50 days for each system. The total computing time would be 250 days for five systems studied. Depending on competition for computational resources, we might be able to run up to 5-8 simulations simultaneously, which totals 30 – 50 days to complete the additional simulations requested for these systems. The subsequent analysis of the trajectories would conservatively take an additional month.

Beyond the long timeframe and significant additional effort that would be required to perform microsecond simulations for the five systems, we believe it is worth highlighting that the MD simulations are not the central focus of this work. The results of the MD simulations were included to provide a potential mechanistic hypothesis with which to understand the observed experimental data. In alignment with the 100 ns MD simulations, the protein interaction data and

PPI assays presented in figure 5 support the hypothesis that the thiazole inhibitors are stabilizing BDK protein levels through alterations in protein dynamics and protein-protein interactions. Microsecond timescale MD simulations will not change these experimental data nor will they fundamentally change the conclusions presented in this work.

We have added an acknowledgement of the limitation of the timescale of our MD simulations to the Discussion section indicating that not running on the microsecond timescale may potentially lead to missing wider conformational changes. Hopefully this response addresses the points raised by Reviewer 3 and explains why the request, though academically interesting, is an ask that will be very challenging to accomplish in a reasonable timeframe and arguably will provide minor additional benefit to the overall conclusions of the manuscript.

We again thank reviewer 3 for their time and interest in our work. We hope that our manuscript will now be acceptable for publication in Nature Communications with these additional clarifications.

Best Regards,

Rachel J. Roth Flach, PhD
Associate Research Fellow
Internal Medicine Research Unit
Pfizer, Inc
Cambridge, MA 02139

Kevin J. Filipski
Associate Research Fellow
Medicine Design
Pfizer, Inc
Cambridge, MA 02139